# FEATURE RECONSTRUCTION FROM OUTPUTS CAN MITIGATE SIMPLICITY BIAS IN NEURAL NETWORKS

**Sravanti Addepalli**[†][◇][*]    **Anshul Nasery**[†][*]    **Praneeth Netrapalli**[†]
**Venkatesh Babu R.**[◇]    **Prateek Jain**[†]
[†] Google Research India [◇] Indian Institute of Science, Bangalore

## ABSTRACT

Deep Neural Networks are known to be brittle to even minor distribution shifts compared to the training distribution. While one line of work has demonstrated that *Simplicity Bias* (SB) of DNNs – bias towards learning only the simplest features – is a key reason for this brittleness, another recent line of work has surprisingly found that diverse/ complex features are indeed learned by the backbone, and their brittleness is due to the linear classification head relying primarily on the simplest features. To bridge the gap between these two lines of work, we first hypothesize and verify that while SB may not altogether preclude learning complex features, it amplifies simpler features over complex ones. Namely, simple features are replicated several times in the learned representations while complex features might not be replicated. This phenomenon, we term *Feature Replication Hypothesis*, coupled with the *Implicit Bias* of SGD to converge to maximum margin solutions in the feature space, leads the models to rely mostly on the simple features for classification. To mitigate this bias, we propose *Feature Reconstruction Regularizer (FRR)* to ensure that the learned features can be reconstructed back from the logits. The use of *FRR* in linear layer training (*FRR-L*) encourages the use of more diverse features for classification. We further propose to finetune the full network by freezing the weights of the linear layer trained using *FRR-L*, to refine the learned features, making them more suitable for classification. Using this simple solution, we demonstrate up to 15% gains in OOD accuracy on the recently introduced semi-synthetic datasets with extreme distribution shifts. Moreover, we demonstrate noteworthy gains over existing SOTA methods on the standard OOD benchmark DomainBed as well.

## 1 INTRODUCTION

Despite the remarkable success of Deep Neural Networks (DNNs) in various fields, they are known to be brittle against even minor shifts in the data distribution during inference, which are not uncommon in a real world setting (Quinonero-Candela et al., 2008; Torralba & Efros, 2011). For example, a self-driving car that works well in normal weather may perform poorly when it is snowing, leading to disastrous outcomes. The need for improving the robustness of such systems against distribution shifts has sparked interest in the area of Out-Of-Distribution or OOD generalization (Hendrycks & Dietterich, 2019; Gulrajani & Lopez-Paz, 2020).

In this work, we aim to tackle the problem of OOD generalization of Neural Networks in a covariate-shift (Shimodaira, 2000) based classification setting, by addressing the fundamental cause of their brittleness, rather than by explicitly enforcing invariances in the network using domain labels or data augmentations. More specifically, we aim to mitigate the issue of *Simplicity Bias*, which is the tendency of *Stochastic Gradient Descent (SGD)* based solutions to overly rely on simple features alone, rather than on a diverse set of features (Arpit et al., 2017; Valle-Perez et al., 2018). While this behavior was earlier used to explain the remarkable generalization of Deep Networks, recent works suggest that this is indeed a key reason behind their brittleness to domain shifts (Shah et al., 2020).

---

[*]Equal Contribution. Correspondence to {sravantia, anshulnasery}@google.com

The extent of Simplicity Bias seen in models is a result of two important factors - diversity of features learned by the feature extractor[1], and the extent to which these diverse features are used for the task at hand, such as classification[2]. Recent works suggest that generalization to distribution shifts can be improved by retraining the last layer alone, indicating that the features learned may already be good enough for the same (Rosenfeld et al., 2022; Kirichenko et al., 2022b). Does this imply that brittleness of models can be attributed to the learning of the classification head alone? If this is the case, why does SGD fail to utilize these diverse features despite its *Implicit Bias* to converge to a maximum margin solution in a linearly separable case (Soudry et al., 2018)? To answer these questions, we firstly hypothesize and empirically verify that Simplicity Bias leads to the learning of simple features over and over again, as compared to other, more complex features. For example, among the $512$ penultimate layer features of a ResNet, $462$ of them might capture a simple feature such as *color*, while the remaining $50$ might capture a more complex feature such as *shape* – we refer to this as *(Simple) Feature Replication Hypothesis*. Assuming feature replication hypothesis, we further show theoretically and empirically that a maximum margin classifier in the replicated feature space would give much higher importance to the replicated feature when compared to others, highlighting why the linear layer relies more on simpler features for classification.

To mitigate this, we propose a novel regularizer termed *Feature Reconstruction Regularizer (FRR)*, to enforce that the features learned by the network can be reconstructed back from the logit or pre-softmax layer used for the classification task. As shown in Fig.2, we firstly propose to train the linear classifier alone by freezing the weights of the feature extractor. This formulation enables the learning of an *Invertible Mapping* in the output layer, specifically for the domain of features seen during training. This further allows the logit layer to act as an information bottleneck, encouraging all the factors of variation in the features to be utilized for the classification task, thereby improving the diversity of features used. We theoretically show that adding this constraint while finetuning the linear layer can learn a max-margin classifier in the original input space, disregarding feature replication. Consequently, the learnt linear classifier also gives more importance to non replicated complex features while making predictions. We further explore the possibility of improving the quality of features learned by the feature extractor, by using *FRR* for finetuning the backbone as well. In order to do this, we freeze the linear classification head, and further finetune the backbone with FRR. We find that this encourages the network to indeed learn better quality features that are more relevant for classification. We list the key contributions of this work below -

- *Key Observation*: We provide a crisp hypothesis of "feature replication" to explain the brittleness of ERM trained neural networks to OOD data (Sec 3.1). Using this, we further provide theoretical and empirical evidence to justify the existence of Simplicity Bias in maximum margin classifiers.
- *Novel Algorithm based on the Observation*: Based on this, we introduce a novel FRR regularizer to safeguard against the feature replication phenomenon (Sec 3.2). We also provide theoretical support for FRR in an intuitive data distribution setting. Furthermore, we introduce a simple *FRR-L* method to only regularize the linear head with FRR, and then introduce *FRR-FLFT* training regimen to train the feature extractor for improved OOD robustness (Sec 4).
- *Empirical validation of the hypothesis and the proposed algorithm*: We demonstrate the effectiveness of FRR-FLFT and FRR-L by conducting extensive experiments on semi-real datasets (Table 2) constructed to study OOD brittleness, as well as on standard OOD generalization benchmarks, where FRR-FLFT can provide up to 3% gains over SOTA methods for OOD generalization(Table 3).

## 2    RELATED WORKS

**Learning diverse classifiers to counter simplicity bias:** Recent works have shown that ERM trained models learn diverse features, however, the linear layer fails at capturing and utilizing these diverse features properly. There have been several attempts at training classifiers which can make use of such diverse features. Teney et al. (2022) train a number of linear classifiers on top of a pre-trained network with a diversity regularizer, which encourages the classifiers to rely on different features. Xu et al. (2022) and Bahng et al. (2020) propose to train debiased classifiers which are statistically independent from trained biased networks, but these need careful design and prior knowledge of the

---

[1]In this paper, we refer to the penultimate layer's activations as *features*.

[2]i.e., by the final classification layer.

biases in trained networks. Kirichenko et al. (2022a) show that reweighting train set examples and retraining the last layer of a pre-trained deep network can alleviate spurious correlations, provided one can access a balanced dataset. In contrast to these methods, our method can work simply on the training set data, and produce a single classifier which is debiased. Huang et al. (2020) propose to mute the features with highest gradients, and use only the other features to make a prediction. While this method suppresses the maximally used features, it does not encourage the learning of hard-to-learn features, which is directly realized using our loss formulation. Kumar et al. (2022) suggest that finetuning the final linear layer first before finetuning the entire network can make it more robust to OOD shifts, and we utilize this insight in the FRR-FLFT phase of our method. A complementary approach to this problem is to learn features that are more diverse (Zhang et al., 2022; Wang et al., 2019). We note that applying our proposed method on top of such techniques would encourage the classifier to use the diverse features effectively, and this can further benefit the performance.

**Domain Generalization and OOD robustness:** The performance of neural networks is known to drop when there is a mismatch in the train and test distributions (Hendrycks & Dietterich, 2019), and methods to mitigate this have been gaining a lot of attention in recent years. The problem has been studied under various assumptions on distribution shift. The commonly studied setting of domain generalization (Gulrajani & Lopez-Paz, 2020; Li et al., 2018a) assumes that the train distribution consists of a mixture of distinct distributions (called domains), with each train sample having a domain label associated with it. The stronger setting of aggregate domain generalization (Thomas et al., 2021; Matsuura & Harada, 2020) assumes training data to be drawn from a mixture of distributions, but does not assume the availability of domain labels. Finally, OOD robustness (Hendrycks & Dietterich, 2019; Koh et al., 2021) drops all of these assumptions. Most works tackling the domain generalization problem attempt to train a model whose predictions are invariant to the domain label (Li et al., 2018a; Arjovsky et al., 2019), or try to align the features of the model for examples from different domains (Shi et al., 2021; Shankar et al., 2018). However, since we aim to tackle the stronger setting of OOD generalization, we do not use domain labels. Tackling the OOD robustness problem, Thomas et al. (2021) and Matsuura & Harada (2020) first cluster training examples into "pseudo-domains", after which standard domain generalization techniques are used. Another recent line of works propose using model averaging (Cha et al., 2021; Li et al., 2022) and/or ensembling (Arpit et al., 2021) for better OOD generalization. These techniques are complementary to our contribution, and we demonstrate how they can benefit each other in our empirical evaluation.

## 3 FEATURE REPLICATION HYPOTHESIS

Prior works have shown that neural networks trained with SGD exhibit simplicity bias (SB), even when initialized with pre-trained models that can capture complex features. Our Feature Replication Hypothesis – FRH– states that: SB is observed because the simpler features of the input are *replicated* multiple times in the feature space of neural networks.

When trained using SGD, the final linear layer then learns the max margin classifier on these replicated features, which leads to over-reliance on simpler features in the input. Hence, the outputs of the network are brittle to distribution shifts that change such replicated features. In this section, we provide empirical and theoretical evidence for FRH, and propose a new regularizer – FRR– to mitigate this effect.

We first introduce some useful notations. Let $f_\theta(x) : \mathbb{R}^d \to \mathbb{R}^m$ be the feature extractor of a neural network parameterized by weights $\theta$, and $W \in \mathbb{R}^{m \times k}$ be the weight matrix of the linear classifier. For input $x \in \mathbb{R}^d$, the output of the network is $W^T f_\theta(x) \in \mathbb{R}^k$.

### 3.1 EMPIRICAL VALIDATION OF FEATURE REPLICATION HYPOTHESIS (FRH) IN ERM

**Coloured MNIST dataset.** To empirically demonstrate feature replication, we use a binarized version of the coloured MNIST dataset (Gulrajani & Lopez-Paz, 2020). We construct this dataset by first assigning two digits of the MNIST dataset, namely "1" and "5", to classes 0 and 1 respectively. While training the network, we super-impose images of "1" onto colours of range $R_0 = [(115, 0, 0) - (256, 141, 0)]$ (i.e. red), and images of "5" onto colours of range $R_1 = [(0, 115, 0) - (141, 256, 0)]$ (i.e. green). The dataset is constructed such that the simple feature, namely colour, is weakly correlated with the labels, while the complex shape features are strongly correlated with labels. See Appendix D.1 for more details about the dataset.

Table 1: **Features replication in Coloured MNIST**: We observe that ERM learns more colour features than shape features, and the prediction is less correlated with the shape features. Adding FRR makes the network depend more on shape and less on colour, leading to better OOD performance.

| Algorithm | Number | | Average correlation with input | | Correlation with output | | ID Accuracy | OOD Accuracy |
|---|---|---|---|---|---|---|---|---|
| | Colour | Shape | Colour | Shape | Colour | Shape | | |
| ERM | 26 | 4 | 0.76 | 0.26 | 0.81 | 0.61 | 99.9% | 59.1% |
| ERM+FRR-L | 26 | 4 | 0.76 | 0.26 | 0.71 | 0.65 | 99.6% | 64.9% |

**Training setup**: We train a model on this dataset, and test it on images which do not have any correlation between the label and the colour, i.e. images where the digits "1" and "5" are superimposed on randomly coloured backgrounds. We construct this test distribution to see how well different algorithms learn simple (i.e. colour) and complex (i.e. shape) features, since an algorithm which depends only on the spurious colour features would not have good performance on the test domain. We train a four layered CNN on this data. If a feature in the penultimate layer $f_\theta(x)$ has more than $90\%$ correlation with the color or shape of the input, then we call it as a color feature or a shape feature, respectively. We also compute the correlation of these features with the output of the network ($W^T f_\theta(x)$) over inputs from the test domain. This gives us information of the learnt features, and their contributions to the final prediction of the network. Note that the feature dimension is $m = 32$, and the output dimension is $k = 1$.

**Observations**: In Table 1, we report the number of colour features, shape features, and the average correlation of each of these with the final prediction. We observe that the ERM trained model learns both shape and colour features, but the number of learnt colour features (26) is much higher than the number of shape features (4), despite their weaker correlation with labels, thus validating our *Feature Replication Hypothesis*. We also visualize the inter-feature correlation of the learnt features in Fig 5, which shows blocks of highly correlated features, further validating our hypothesis. We note that correlation of the output with the shape features is lower, leading to OOD accuracy of $59\%$.

## 3.2 Feature Reconstruction Regularizer (FRR)

To alleviate simple feature replication issue, we propose *Feature Reconstruction Regularizer* (FRR) to enforce that the learned features can be reconstructed from the output logits. We propose to retrain the final linear layer using this regularizer to allow the model to utilize diverse features to compute the final output. We implement this by introducing another neural network with the objective of reconstructing the features of the network from the output logits, i.e. features $f_\theta(x)$ should be recoverable from the predictions of the network through a transform $\mathcal{T}_\phi(.)$ parameterized by $\phi$. That is, FRR is given by:

$$\mathcal{L}_{\text{FRR}}(x, \theta, W, \phi) = ||f_\theta(x) - \mathcal{T}_\phi(W^T f_\theta(x))||_p \tag{1}$$

where $||.||_p$ denotes the $\ell_p$ norm. We set this norm to be $\ell_\infty$ or $\ell_1$ in our experiments. In the simplest case, $\mathcal{T}_\phi(y) = \phi y$, where $\phi \in \mathbb{R}^{m \times k}$. Note that in order to find the appropriate $\phi$, we jointly optimize $W$ and $\phi$ using gradient descent based optimizers. We also experiment with $\phi$ being a more complex neural network.

We **empirically** validate FRR on **Coloured MNIST**, where using FRR with the linear layer leads to lower correlation with Colour compared to standard ERM (Table 1). Consequently, OOD accuracy improves by 5% over ERM.

## 3.3 FRH & FRR: Theoretical Analysis

We now present a simple and intuitive data distribution with feature replication that highlights the OOD brittleness of standard ERM, and also demonstrates FRR can be significantly more robust.

**Data Distribution**: Consider a linearly separable distribution consisting of two factors of variation as shown in Figure 1. That is, consider the following distribution $(x, y) \sim \mathcal{D}$, where,

$$y = \pm 1 \text{ with probability } 0.5, \quad x = [y, \ y] + [n_1, \ n_2] \in \mathbb{R}^2, \quad n_i \sim \text{Unif}[-0.5, 0.5], i \in [2]. \tag{2}$$

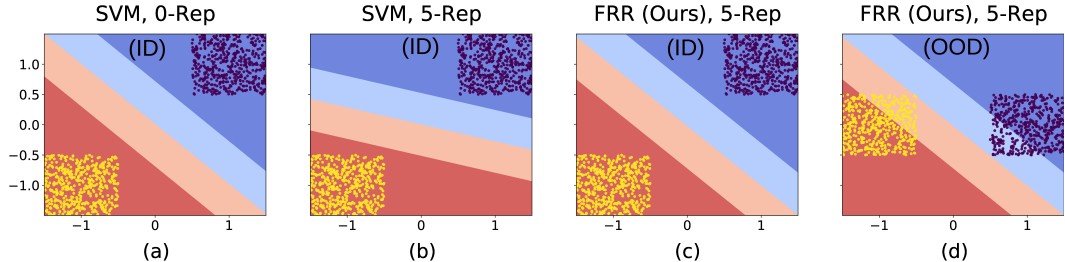

Figure 1: We demonstrate the brittleness of SVM (a, b) and effectiveness of FRR (c, d) based classifiers on a toy dataset comprising of 2 factors of variation, sampled from a uniform distribution. We consider $d$ (= 0 or 5) feature replications (Rep) along the y-axis. FRR converges to a maximum margin solution in the non-replicated feature space, resulting in improved OOD robustness (d)

Also consider a feature extractor $f_\theta(.)$ which captures feature replication in the first feature, i.e. for every data point $(x, y)$, the new, feature replicated data point will be $(\tilde{x}, y)$, where,

$$f_\theta(x) = \tilde{x} = [x_1, \cdots, x_1, x_2] \in \mathbb{R}^{d+1}, \tag{3}$$

i.e., $x_1$ is repeated $d$ times. The joint distribution of features and labels is denoted by $\tilde{\mathcal{D}}$. Finally, we define the $l_2$ max margin classifier over a distribution $\mathcal{D}$ as $w_{\text{MM}} := \arg\min_w \frac{1}{2} \|w\|_2^2$ subject to $y \cdot \langle w, x \rangle \geq 1 \ \forall \ (x, y) \in \text{Supp}(\mathcal{D})$. Then we have the following results:

**Claim 3.1** (Brittleness due to Feature Replication). *Consider the data distribution given in Equation 2, 3. Then, the following holds: (1.) The max-margin classifier $w_{MM}$ over $\mathcal{D}$ is given by $w_{MM} = [1, 1]$, and (2.) The max-margin classifier $\tilde{w}_{MM}$ over $\tilde{\mathcal{D}}$ is given by $\tilde{w}_{MM} = \left[\frac{2}{d+1}, \cdots, \frac{2}{d+1}\right] \in \mathbb{R}^{d+1}$.*

The above claim implies that when there are replicated features to the input of the linear layer, the max-margin classifier would give much more importance to the feature that is replicated. Hence, even a minor change in this replicated feature in the input space would be *amplified* in the output of the classifier. This is especially concerning in light of the observations in Table 1, which validate the Feature Replication Hypothesis in Coloured MNIST.

**Proposition 3.2** (Robustness of FRR). *Denote the average feature reconstruction loss $\mathcal{L}_{FRR}(\tilde{w}, \tilde{\phi}) := \max_{1 \leq i \leq d+1} \mathbb{E}_{(\tilde{x}, y) \sim \tilde{\mathcal{D}}} \left[ (\langle \tilde{w}, \tilde{x} \rangle \tilde{\phi}_i - \tilde{x}_i)^2 \right]$ and consider any $(\tilde{w}^*, \tilde{\phi}^*)$ satisfying:*

$$(\tilde{w}^*, \tilde{\phi}^*) \in \arg\min_{(\tilde{w}, \tilde{\phi})} \mathcal{L}_{FRR}(\tilde{w}, \tilde{\phi}) \text{ subject to } y \cdot \langle \tilde{w}, \tilde{x} \rangle \geq 0 \ \forall \ (\tilde{x}, y) \in \text{Supp}\left(\tilde{\mathcal{D}}\right).$$

*We have that: $\tilde{w}_1^* + \cdots + \tilde{w}_d^* = \tilde{w}_{d+1}^*$. Consequently, we have $\langle \tilde{w}^*, \tilde{x} \rangle \propto \langle w_{MM}, x \rangle$ for all $x \in \mathbb{R}^2$.*

Practically, we can implement the above as $\ell_{2,\infty}$ over a batch. Above result shows that the feature reconstruction regularizer will produce a linear classifier that gives equal weights to the replicated and non-replicated features. This is equivalent to a maximum margin classifier in the *non-replicated feature space*, thereby resulting in enhanced robustness to distribution shifts. Same is reflected in Figure 1 (c), (d) which show impact of FRR on the trained boundary in the non-replicated feature space. We defer the proofs of the above to Appendix A. We also provide a more general result by assuming *correlated* feature representations in Appendix A.

## 4 TRAINING PROCEDURE

**Pretraining** : In order to learn features which are relevant to the train distribution, we first pretrain our model using standard ERM with the cross-entropy loss $\mathcal{L}_{std}(W, \theta, (x, y))$.

**FRR-L** : Since ERM training is known to learn several rich and diverse features, we freeze the backbone parameters $\theta$, and retrain the final layer $W$ as the following-

$$(W_{\text{FRR}}, \phi_{\text{FRR}}) = \min_{W, \phi} \mathcal{L}_{\text{std}}(W, \theta, (x, y)) + \lambda_{\text{L}} \mathcal{L}_{\text{FRR}}(x, \theta, W, \phi) \tag{4}$$

where $\lambda_L$ is a hyperparameter weighing the two losses. We train $W$ and $\phi$ jointly. We refer to this step as FRR-L, i.e. Feature Reconstruction Regularizer - Linear, since we only train the linear layer.

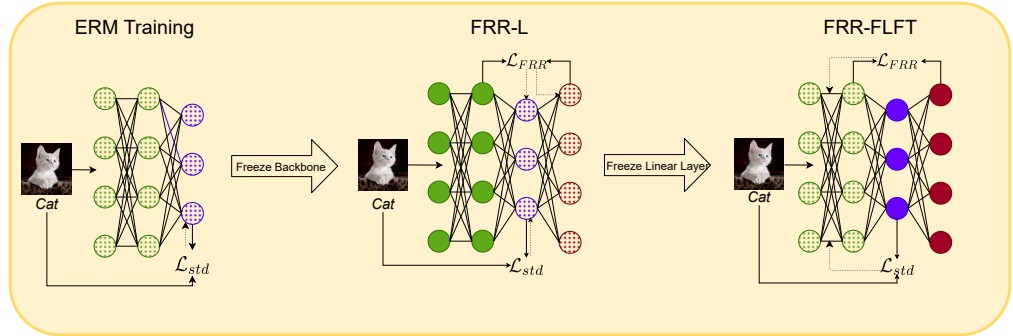

Figure 2: **Our training procedure**: Dotted fill indicates that the parameters are trainable.

**FRR-FLFT** : Following the suggestions of Kumar et al. (2022), we follow up the linear layer training with the finetuning of the feature extractor $\theta$ with a weighted combination of the cross-entropy loss and FRR, weighted by a hyper-parameter $\lambda_{FLFT}$. In this step we freeze the weights of the linear layer to improve the stability of training. We do this since naively using this constraint during network training could amplify the Simplicity Bias in networks in degenerate cases. For example, the backbone could learn to output a single replicated simple feature which is predictive enough on the training data. Reconstructing such a feature from logits would also be easy, but such a network might not generalize well. Formally, the optimization problem for this step is -

$$\theta_{\text{FLFT}} = \min_{\theta} \ \mathcal{L}_{\text{std}}(W_{\text{FRR}}, \theta, (x, y)) + \lambda_{\text{FLFT}} \mathcal{L}_{\text{FRR}}(x, \theta, W_{\text{FRR}}, \phi_{\text{FRR}}) \tag{5}$$

We view this step as "sharpening" the features for more accurate predictions. Freezing the linear head makes sure that the features do not collapse to a degenerate solution. Our training algorithm is summarized in Algorithm-1 and the training pipeline is illustrated in Figure 2.

## 5 EXPERIMENTAL RESULTS

### 5.1 UNDERSTANDING HOW FRR MITIGATES SIMPLICITY BIAS

To empirically illustrate the extent of Simplicity Bias in Neural Networks, Shah et al. (2020) introduced several synthetic and semi-synthetic datasets, where some features are explicitly *simple*, requiring a simpler decision boundary for prediction; while others are *complex*. In this section, we demonstrate the effectiveness of the proposed *Feature Reconstruction Regularizer* towards mitigating Simplicity Bias, by evaluating the same on a 10-class variant of the proposed semi-synthetic MNIST-CIFAR dataset, as discussed in the following section.

### 5.1.1 MNIST-CIFAR-10 DATASET

We extend the simple binary MNIST-CIFAR dataset proposed by Shah et al. (2020) to a 10-class dataset, in order to evaluate the impact of the proposed *Feature Reconstruction Regularizer* in a more complex scenario when compared to the binary Colored-MNIST dataset presented in Section-3. We refer to this dataset as MNIST-CIFAR-10. The higher complexity of this dataset allows for a more reliable evaluation of various settings such as linear probing, full network finetuning and fixed-linear finetuning, with better granularity of results.

To construct this dataset, we first define correspondences between the classes of CIFAR-10 and MNIST. Each image from class $k$ of MNIST is mapped with an image from class $k$ of CIFAR-10, with the label being set to $k$. Thus, every training data sample $(x_1, x_2, y)$ consists of $x_1$ and $x_2$, which are images from CIFAR-10 and MNIST respectively, along with their ground truth class $y$. It is to be noted that for both CIFAR-10 and MNIST, labels are always correlated with the respective images. In such a scenario, although a classifier can achieve very good performance by relying solely on the simple (MNIST) features, the goal of Out-Of-Distribution (OOD) robustness requires it to rely on the complex (CIFAR-10) features as well. This dataset represents the toughest setting of OOD generalization, where there is no differentiation between important features and spurious

Table 2: ID and OOD accuracy (%) by training on MNIST-CIFAR-10 in various training regimes.

| Initialization | Layers trained | Exp ID | Training Loss | Training Dataset | In-Distribution (ID) MNIST-CIFAR-10 | Out-Of-Distribution (OOD) MNIST-AvgCIFAR | CIFAR-AvgMNIST |
|---|---|---|---|---|---|---|---|
| Random | All layers | E1 | M1: ERM (CE) | MNIST-CIFAR-10 | $99.84_{\pm 0.01}$ | $97.44_{\pm 0.89}$ | $51.92_{\pm 1.52}$ |
| | | E2 | Cross-Entropy | CIFAR-RandMNIST | $88.53_{\pm 0.15}$ | $9.77_{\pm 0.11}$ | $88.52_{\pm 0.14}$ |
| | | E3 | Cross-Entropy | MNIST-RandCIFAR | $99.68_{\pm 0.02}$ | $94.84_{\pm 1.18}$ | $10.02_{\pm 0.13}$ |
| M1:ERM | Linear layer | E4 | M2: ERM-L (CE) | MNIST-CIFAR-10 | $99.86_{\pm 0.01}$ | $97.06_{\pm 0.05}$ | $52.73_{\pm 0.08}$ |
| | | E5 | Cross-Entropy | CIFAR-RandMNIST | $65.14_{\pm 0.05}$ | $10.15_{\pm 0.01}$ | $65.10_{\pm 0.03}$ |
| | | E6 | Cross-Entropy | MNIST-RandCIFAR | $99.71_{\pm 0.00}$ | $94.84_{\pm 0.04}$ | $10.33_{\pm 0.17}$ |
| | | E7 | CE + Full-Rank Reg | MNIST-CIFAR-10 | $99.86_{\pm 0.01}$ | $97.04_{\pm 0.15}$ | $52.87_{\pm 1.09}$ |
| | | E8 | M3: FRR-L (**Ours**) | MNIST-CIFAR-10 | $99.88_{\pm 0.00}$ | $96.81_{\pm 0.38}$ | $59.13_{\pm 0.37}$ |
| M2:ERM-L | All layers | E9 | Cross-Entropy | | $99.84_{\pm 0.02}$ | $97.67_{\pm 0.19}$ | $53.33_{\pm 0.28}$ |
| | Feature extractors | E10 | Cross-Entropy | MNIST-CIFAR-10 | $99.84_{\pm 0.01}$ | $97.67_{\pm 0.18}$ | $53.67_{\pm 0.40}$ |
| | All layers | E11 | FRR-FT | | $99.84_{\pm 0.02}$ | $97.32_{\pm 0.45}$ | $54.12_{\pm 0.44}$ |
| | Feature extractors | E12 | FRR-FLFT | | $99.81_{\pm 0.04}$ | $98.44_{\pm 0.64}$ | $60.02_{\pm 0.69}$ |
| M3:FRR-L | All layers | E13 | Cross-Entropy | | $99.87_{\pm 0.01}$ | $97.03_{\pm 0.35}$ | $61.75_{\pm 0.33}$ |
| | Feature extractors | E14 | Cross-Entropy | MNIST-CIFAR-10 | $99.88_{\pm 0.02}$ | $97.35_{\pm 0.34}$ | $63.73_{\pm 0.62}$ |
| | All layers | E15 | FRR-FT | | $99.85_{\pm 0.01}$ | $99.30_{\pm 0.05}$ | $62.13_{\pm 0.42}$ |
| | Feature extractors | E16 | M4: FRR-FLFT (**Ours**) | | $99.84_{\pm 0.03}$ | $99.45_{\pm 0.03}$ | $68.12_{\pm 0.96}$ |
| M4:FRR-FLFT | Linear layer | E17 | Cross-Entropy | CIFAR-RandMNIST | $79.92_{\pm 0.33}$ | $11.93_{\pm 0.09}$ | $77.35_{\pm 0.10}$ |
| | | E18 | Cross-Entropy | MNIST-RandCIFAR | $99.71_{\pm 0.00}$ | $99.46_{\pm 0.00}$ | $10.27_{\pm 0.12}$ |

correlations. A real-world example of such a case is the classification of swans versus bears, with the training dataset consisting of only white swans and black bears. Here the model could either rely on shape or color for classification. A classification network that relies solely on the simplest feature color, fails to generalize to the test set consisting of black swans and polar bears.

## 5.2 TRAINING AND EVALUATION SETTINGS

We consider two separate ResNet-18 (He et al., 2016) feature extractors for CIFAR-10 and MNIST respectively. The outputs of the Global Average Pooling (GAP) layers in each of the feature extractors are concatenated to form a 1024 dimensional vector, which is given as input to the linear classifier. This architecture allows the computation of accuracy based on either a combination of both CIFAR-10 and MNIST features, or based on features of only one of the datasets. For example, to evaluate the performance of the classifier based on CIFAR-10 features alone, we replace the 512 dimensional MNIST feature vector of each data sample with an average feature vector computed from all images in the MNIST dataset. We refer to this as the CIFAR-AvgMNIST dataset, while the corresponding one for MNIST is refered to as the MNIST-AvgCIFAR dataset. Similar to the work by Shah et al. (2020), we define two additional datasets, CIFAR-RandMNIST and MNIST-RandCIFAR, where images from one of the datasets (MNIST and CIFAR-10 respectively) are randomly shuffled with respect to their corresponding labels. The base training (E1, E2, E3) is done for 500 epochs, and the linear layer training / finetuning (E4 - E18) is done for 20 epochs, without any augmentations.

## 5.3 EXPERIMENTAL RESULTS IN VARIOUS TRAINING REGIMES

We present the results of training on the MNIST-CIFAR-10 dataset using different algorithms in Table 2. The mean and standard deviation across five runs have been reported for each case.

**ERM Training:** By training a randomly initialized model on the MNIST-CIFAR-10 dataset using the cross-entropy loss (E1), we obtain an accuracy of $99.84\%$ on its corresponding test split. While the accuracy of this model on the MNIST-avgCIFAR dataset is high ($97.44\%$), its performance on the CIFAR-avgMNIST dataset is poor ($51.92\%$), indicating that the model chooses to rely more on the simpler MNIST features, rather than a combination of both CIFAR and MNIST features.

While the performance on the CIFAR-avgMNIST and MNIST-avgCIFAR datasets is sufficient to understand the extent of CIFAR/ MNIST features used by the classification head, it does not give a clear picture on the features learned by the two feature extractors. To understand this, we reinitialize the linear classification head randomly, and train the same using CIFAR-RandMNIST (E5) and MNIST-RandCIFAR datasets (E6) respectively. We obtain an accuracy of $65.2\%$ on the CIFAR-avgMNIST dataset in the former case, indicating that although the CIFAR features learned can possibly achieve $13\%$ higher accuracy (w.r.t. E1), the bias in the classification head prevents them from participating in the classification task. The MNIST-avgCIFAR accuracy of the latter case is high as expected. An upper bound on CIFAR-10 and MNIST accuracy that can be achieved with the selected architecture and training strategy (without using any augmentations) can be seen in E2 ($88.53\%$) and E3 ($99.68\%$) respectively.

**Training the Linear Classification Head:** As discussed, while ERM training (E1) learns features that can be used for better OOD performance (E5), it does not effectively leverage these features for the classification task. We firstly explore the possibility of bridging the difference in the CIFAR-avgMNIST accuracy between E1 and E5 by merely retraining the linear layer. By reinitializing and naively retraining the linear layer with Cross-entropy loss, the accuracy on CIFAR-avgMNIST improves by less than $1\%$ (E4). Using the proposed *Feature Reconstruction Regularizer (FRR)* for training the linear layer alone, the CIFAR-avgMNIST accuracy improves by $7.21\%$ as shown in E8, demonstrating the effectiveness of the proposed regularizer in mitigating Simplicity Bias. We penalize the $\ell_\infty$ norm of difference in original features and their reconstruction in addition to the minimization of cross-entropy loss. The reconstruction based regularizer enforces the network to utilize both CIFAR and MNIST features for classification. Since this regularizer resembles an orthonormality constraint on the linear classification head, we additionally check the effectiveness of explicitly enforcing a full-rank constraint on the linear layer by minimizing the following: $||WW^T - I||_F$ (E7). We find that this is not effective in improving the overall accuracy, possibly because it enforces a very stringent constraint on the final classification layer. Contrary to this, the proposed *Feature Reconstruction Regularizer* allows more flexibility by imposing this constraint only on the domain of features seen during training. This accounts for the simple feature replication as well, enabling to view the logit layer as an information bottleneck in the reconstruction.

**Finetuning (FT) and Fixed Linear Finetuning (FLFT):** We explore the finetuning of a given base model in two settings - firstly by finetuning all layers in the network (denoted as FT or FineTuning), and secondly, by freezing the parameters of the linear classification head and finetuning only the feature extractors, which we refer to as FLFT or Fixed Linear FineTuning. By finetuning an ERM trained base model using either of the two strategies (E9 and E10), we observe gains of less than $1\%$. We observe similar gains even by finetuning the full network with FRR (E11). Contrary to this, by using FRR-FLFT even on the ERM trained network (E12), we obtain $7.29\%$ improvement over the base model. This shows that, by allowing the full network to change while imposing the FRR constraint, the network can continue to rely on simple features, possibly by reducing the number of complex features learned by the feature extractor. However, by freezing the weights of the linear layer and further imposing this constraint, the network is forced to refine the CIFAR features that are already being used for prediction.

**Combining FRR-L and FRR-FLFT:** While we obtain similar order of gains ( $\sim 7\%$) using both FRR-L and FRR-FLFT individually, the former improves the diversity of features being considered by the classification head, while the latter improves the quality of the features themselves. We therefore propose a training strategy that combines both FRR-L and FRR-FLFT. Using this, we obtain gains of $16.2\%$ over the ERM baseline as shown in E16, indicating that the combination of FRR-L and FRR-FLFT has a compounding effect by firstly selecting diverse features, and further refining these features to be more useful for classification. Although FRR-L followed by FRR-FT (E15) is also effective, it has about $6\%$ lesser gains when compared to the proposed approach of FRR-L + FRR-FLFT. We note that following up FRR-L with ERM-FT (E13) or ERM-FLFT (E14) also refines the learned features, making them more suitable for the classification task, yielding $2.6\%$ and $4.6\%$ gains respectively over FRR-L.

We verify the quality of features learned by the feature extractors after the proposed training strategy FRR-L + FRR-FLFT by reinitializing and retraining the linear classifier on CIFAR-RandMNIST (E17) and MNIST-RandCIFAR (E18) datasets respectively. We observe considerable gains of around $15\%$ on MNIST-CIFAR-10 accuracy using CIFAR-RandMNIST training when compared to ERM (E5), demonstrating that the proposed approach not only results in more CIFAR features being used for classification, but also leads to the learning of better CIFAR features.

## 5.4 OOD GENERALIZATION IN A REAL WORLD SETTING

We show the efficacy of FRR towards improving OOD generalization on the DomainBed (Gulrajani & Lopez-Paz, 2020) benchmark. We use the performance of the model on in-domain validation data (i.e. the *in-domain* strategy by Gulrajani & Lopez-Paz (2020)) to select the best hyper-parameters, and report the average performance and standard deviation across 5 random seeds.

**Baselines** : We compare our method against standard ERM training, which has proven to be a frustratingly difficult baseline (Gulrajani & Lopez-Paz, 2020), and also against several state of the art methods on this benchmark - SWAD (Cha et al., 2021), MIRO (Cha et al., 2022) and SMA (Arpit

Table 3: Results on DomainBed: The bottom partition shows results of methods that perform model weight averaging. In both cases, with (top) and without (bottom) model weight averaging, the proposed approach outperforms existing methods.

| Algorithm | PACS | VLCS | OfficeHome | TerraIncognita | DomainNet | Avg. |
|---|---|---|---|---|---|---|
| ERM | $85.5_{\pm 0.1}$ | $77.5_{\pm 0.4}$ | $66.5_{\pm 0.2}$ | $46.1_{\pm 0.6}$ | $40.9_{\pm 0.1}$ | 63.3 |
| IRM | $83.5_{\pm 0.8}$ | $78.5_{\pm 0.5}$ | $64.3_{\pm 2.2}$ | $47.6_{\pm 0.8}$ | $33.9_{\pm 2.8}$ | 61.6 |
| CORAL | $86.2_{\pm 0.3}$ | $78.8_{\pm 0.6}$ | $68.7_{\pm 0.3}$ | $47.6_{\pm 1.0}$ | $41.5_{\pm 0.1}$ | 64.5 |
| MIRO | $85.4_{\pm 0.4}$ | $79.0_{\pm 0.0}$ | $70.5_{\pm 0.4}$ | $50.4_{\pm 1.1}$ | $44.3_{\pm 0.2}$ | 65.9 |
| ERM+FRR-L | $85.7_{\pm 0.1}$ | $76.6_{\pm 0.2}$ | $68.4_{\pm 0.2}$ | $53.7_{\pm 0.6}$ | $44.2_{\pm 0.1}$ | 65.7 |
| ERM+FRR | $87.5_{\pm 0.1}$ | $77.6_{\pm 0.3}$ | $69.4_{\pm 0.1}$ | $54.1_{\pm 0.6}$ | $45.1_{\pm 0.1}$ | **66.8** |
| SMA | $87.5_{\pm 0.2}$ | $78.2_{\pm 0.2}$ | $70.6_{\pm 0.1}$ | $50.3_{\pm 0.5}$ | $46.0_{\pm 0.1}$ | 66.5 |
| SWAD | $88.1_{\pm 0.1}$ | $79.1_{\pm 0.1}$ | $70.6_{\pm 0.2}$ | $50.0_{\pm 0.3}$ | $46.5_{\pm 0.1}$ | 66.9 |
| SWAD+FRR | $89.2_{\pm 0.4}$ | $80.0_{\pm 0.2}$ | $70.3_{\pm 0.1}$ | $53.2_{\pm 0.3}$ | $46.2_{\pm 0.0}$ | **67.9** |

et al., 2021). Finally, we show that our approach can be effectively integrated with stochastic weight averaging to obtain further gains. See Appendix G for further experimental details.

**Main Results:** The main results of our algorithm are reported in Table 3. We find that our pipeline of training and finetuning with FRR, when combined with ERM achieves improved performance with respect to the state of the art methods that do not use model weight-averaging, and in fact achieves comparable performance to methods that use model weight averaging as well. Further, our method obtains substantial gains of more than 3% over ERM across datasets. The gains are especially pronounced for the larger datasets including DomainNet and TerraIncognita (8% and 5% resp.), indicating the efficacy and scalability of our method. Further, it is clear from Table 3 that finetuning the feature extractor once the linear layer is fixed provides a boost of over 1% on average over FRR-L. This empirically validates our finetuning paradigm which we denote as ERM+FRR. Finally, using our method in tandem with SWAD helps us achieve a new state-of-the-art on the DomainBed benchmark, outperforming other methods on three datasets while achieving comparable performance on two, and being better than existing SOTA by close to 1% on average. We report detailed results and further ablations in Appendices H and J.

## 6 CONCLUSION AND DISCUSSION

In this work, we consider the problem of OOD generalization through the lens of mitigating Simplicity Bias in Neural Network training. To unravel the paradox pertaining to the existence of Simplicity Bias in learning only the simplest features, and the observation that the features learned by large practical models may already be sufficiently diverse, we put forth the *Feature Replication Hypothesis* that conjectures the learning of replicated simple features and sparse complex ones. Combining this with the *Implicit Bias* of SGD to converge to maximum margin solutions, we provide a theoretical justification to the high OOD sensitivity of Neural Networks.

To specifically overcome the effect of simple feature replication in linear layer training, we propose the *Feature Reconstruction Regularizer*, that penalizes the $\ell_p$ norm distance between the features and their reconstruction from the output logits, thus improving the diversity of features used for classification. We further propose to freeze the weights of the linear layer thus trained, and use the FRR regularizer for finetuning the full network, to refine the features to be more useful for the downstream task. We justify the proposed regularizer both theoretically and empirically on synthetic and semi-synthetic datasets, and demonstrate its effectiveness in a real world OOD generalization setting.

We believe and hope that this work can pave the way towards obtaining a better understanding on the underlying causes for OOD brittleness of neural networks, and inspire the development of better algorithms for addressing the same. We believe the proposed regularizer can potentially work effectively in several other settings that involve the use of linear layer training/finetuning, such as domain adaptation and transfer learning. While the regularizer works effectively in a scenario where the network is first trained using an algorithm such as ERM to learn features that are *relevant* to the task at hand, the robustness of the proposed algorithm in the presence of severely non-relevant features is yet to be explored.

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

## A    PROOFS OF THEORETICAL RESULTS

In this section, we present a generalization of Claim 3.1 and Proposition 3.2 under weaker assumptions on the featurizer, and use them to prove the claims made in Sec 3.

**A more general setting**    Consider the dataset distribution $\mathcal{D}$ as defined in eq equation 2. Now, let $f_\Theta(x) = \Theta x$, where $\Theta \in^{d \times 2}$. Further, $\forall i \, \theta_{i,1} + \theta_{i,2} = 1$. In simple words, the feature extractor maps the input to a $d$-dimensional feature representation, where each feature is a convex combination of the input feature variations. Also note that $\theta = \begin{bmatrix} 1 & 0 \\ 1 & 0 \\ \vdots & \vdots \\ 1 & 0 \\ 0 & 1 \end{bmatrix}$ corresponds to feature replication.

We now rephrase the results from Sec 3.3 in this setting and provide proofs for the same.

**Claim A.1.** *(Restating Claim 3.1) - The max margin classifier $\boldsymbol{w}$ minimizing $\|\boldsymbol{w}\|^2$ and satisfying $y \langle \boldsymbol{w}, \Theta \boldsymbol{x} \rangle \geq 1$ is given by $\boldsymbol{w} = \left[ \frac{2}{d}, \cdots, \frac{2}{d} \right]$.*

*Proof.* Consider the point $\boldsymbol{x} = (0.5, 0.5)$. Then, due to the constraint on the max-margin classifier, we have $y \langle \boldsymbol{w}, \Theta \boldsymbol{x} \rangle \geq 1$ i.e.

$$\frac{1}{2} \sum_{j=1}^{d} w_j (\theta_{j,1} + \theta_{j,2}) \geq 1$$

$$\frac{1}{2} \sum_{j=1}^{d} w_j \geq 1$$

The minimizer of $\ell_2$ norm under this constraint would be when all $w_j = \frac{2}{d}$ for all $j$.

$\square$

Note that the "effective classifier" $\tilde{w}$ in the input space in this case is $\langle w, \Theta \rangle$, i.e. the slope of the classifier in the input space $\frac{\tilde{w}_2}{\tilde{w}_1} = \frac{\sum_{j=1}^{d} \theta_{j,2}}{\sum_{j=1}^{d} \theta_{j,1}}$. In particular, for the case of feature replication, $\sum_{j=1}^{d} \theta_{j,2} = d - 1$ and $\sum_{j=1}^{d} \theta_{j,1} = 1$, leading to a skewed classifier.

Now we restate and show the robustness of FRR in this setting.

**Proposition A.2.** *(Restating Prop 3.2) - Denote the feature reconstruction regularizer for this setting as -*

$$FRR(\boldsymbol{w}) = \min_{U} \max_{1 \leq i \leq d} \mathbb{E}[(\langle \boldsymbol{w}, \Theta \boldsymbol{x} \rangle u_i - (\Theta \boldsymbol{x})_i)^2]$$

*Let $\boldsymbol{w}_{FRR}$ be the minimizer of $FRR(\boldsymbol{w})$ satisfying $y \langle \boldsymbol{w}_{FRR}, \Theta \boldsymbol{x} \rangle \geq 0$ (i.e. it is a perfect classifier). Then, $\boldsymbol{w}_{FRR}$ satisfies*

$$\frac{\langle \boldsymbol{w}_{FRR}, \Theta_{\cdot, 2} \rangle}{\langle \boldsymbol{w}_{FRR}, \Theta_{\cdot, 1} \rangle} = \frac{1 - (\theta_{b,1} - \theta_{a,2})}{1 + (\theta_{b,1} - \theta_{a,2})}$$

*where $a = \arg \max_i \theta_{i,2}$, $b = \arg \max_i \theta_{i,1}$.*

*Proof.* Consider the FRR for this dataset -

$$FRR(\boldsymbol{w}) = \min_{U} \max_{1 \leq i \leq d} \mathbb{E}[(\langle \boldsymbol{w}, \Theta \boldsymbol{x} \rangle u_i - (\Theta \boldsymbol{x})_i)^2]$$

For each $i$, by computing the minimum over $u_i$ by considering the quadratic in $u_i$, we obtain

$$FRR(\boldsymbol{w})_i = \mathbb{E}[(\Theta\boldsymbol{x})_i^2] - \frac{\mathbb{E}[\langle\boldsymbol{w}, \Theta\boldsymbol{x}\rangle(\Theta\boldsymbol{x})_i]^2}{\mathbb{E}[\langle\boldsymbol{w}, \Theta\boldsymbol{x}\rangle^2]}$$

We now consider each term in this expression

$$\mathbb{E}[(\Theta\boldsymbol{x})_i^2] = \mathbb{E}[(\theta_{i,1}x_1 + \theta_{i,2}x_2)^2]$$
$$= \frac{13}{12}(\theta_{i,1}^2 + \theta_{i,2}^2) + 2\theta_{i,1}\theta_{i,2}$$

This is because $\mathbb{E}[x_1x_2] = 1$ and $\mathbb{E}[x_1^2] = \mathbb{E}[x_2^2] = 1 + \frac{1}{12}$.

$$\mathbb{E}[\langle\boldsymbol{w}, \Theta\boldsymbol{x}\rangle(\Theta\boldsymbol{x})_i]^2 = \mathbb{E}[(x_1\langle\boldsymbol{w}, \Theta_{\cdot,1}\rangle + x_2\langle\boldsymbol{w}, \Theta_{\cdot,2}\rangle)(\theta_{1,i}x_1 + \theta_{2,i}x_2)]^2$$
$$= (\mathbb{E}[(x_1^2\langle\boldsymbol{w}, \Theta_{\cdot,1}\rangle\theta_{i,1} + x_2^2\langle\boldsymbol{w}, \Theta_{\cdot,2}\rangle\theta_{i,2} + (x_1x_2)(\langle\boldsymbol{w}, \Theta_{\cdot,1}\rangle\theta_{i,2} + \langle\boldsymbol{w}, \Theta_{\cdot,2}\rangle\theta_{i,1})]))^2$$
$$= (\frac{13}{12}(\langle\boldsymbol{w}, \Theta_{\cdot,1}\rangle\theta_{i,1} + \langle\boldsymbol{w}, \Theta_{\cdot,2}\rangle\theta_{i,2}) + (\langle\boldsymbol{w}, \Theta_{\cdot,1}\rangle\theta_{i,2} + \langle\boldsymbol{w}, \Theta_{\cdot,2}\rangle\theta_{i,1}))^2$$
$$= \frac{13^2}{12^2}(\langle\boldsymbol{w}, \Theta_{\cdot,1}\rangle^2\theta_{i,1}^2 + \langle\boldsymbol{w}, \Theta_{\cdot,2}\rangle^2\theta_{i,2}^2 + 2\langle\boldsymbol{w}, \Theta_{\cdot,2}\rangle\langle\boldsymbol{w}, \Theta_{\cdot,1}\rangle\theta_{i,2}\theta_{i,1})$$
$$+ \langle\boldsymbol{w}, \Theta_{\cdot,1}\rangle^2\theta_{i,2}^2 + \langle\boldsymbol{w}, \Theta_{\cdot,2}\rangle^2\theta_{i,1}^2 + 2\langle\boldsymbol{w}, \Theta_{\cdot,2}\rangle\langle\boldsymbol{w}, \Theta_{\cdot,1}\rangle\theta_{i,1}\theta_{i,2}$$
$$+ \frac{13}{6}((\langle\boldsymbol{w}, \Theta_{\cdot,1}\rangle^2 + \langle\boldsymbol{w}, \Theta_{\cdot,2}\rangle^2)\theta_{i,1}\theta_{i,2} + (\langle\boldsymbol{w}, \Theta_{\cdot,1}\rangle\langle\boldsymbol{w}, \Theta_{\cdot,2}\rangle)(\theta_{i,1}^2 + \theta_{i,2}^2))$$

Finally,

$$\mathbb{E}[\langle\boldsymbol{w}, \Theta\boldsymbol{x}\rangle^2] = \mathbb{E}[(x_1\langle\boldsymbol{w}, \Theta_{\cdot,1}\rangle + x_2\langle\boldsymbol{w}, \Theta_{\cdot,2}\rangle)^2]$$
$$= \frac{13}{12}(\langle\boldsymbol{w}, \Theta_{\cdot,1}\rangle^2 + \langle\boldsymbol{w}, \Theta_{\cdot,2}\rangle^2) + 2\langle\boldsymbol{w}, \Theta_{\cdot,1}\rangle\langle\boldsymbol{w}, \Theta_{\cdot,2}\rangle$$

Putting it together,

$$FRR(\boldsymbol{w})_i = \frac{(\frac{13^2}{12^2} - 1)(\langle\boldsymbol{w}, \Theta_{\cdot,1}\rangle\theta_{i,2} - \langle\boldsymbol{w}, \Theta_{\cdot,2}\rangle\theta_{i,1})^2}{\frac{13}{12}(\langle\boldsymbol{w}, \Theta_{\cdot,1}\rangle^2 + \langle\boldsymbol{w}, \Theta_{\cdot,2}\rangle^2) + 2\langle\boldsymbol{w}, \Theta_{\cdot,1}\rangle\langle\boldsymbol{w}, \Theta_{\cdot,2}\rangle}$$
$$= \frac{(\frac{13^2}{12^2} - 1)(\langle\boldsymbol{w}, \Theta_{\cdot,1}\rangle - (\langle\boldsymbol{w}, \Theta_{\cdot,1}\rangle + \langle\boldsymbol{w}, \Theta_{\cdot,2}\rangle)\theta_{i,1})^2}{\frac{13}{12}(\langle\boldsymbol{w}, \Theta_{\cdot,1}\rangle^2 + \langle\boldsymbol{w}, \Theta_{\cdot,2}\rangle^2) + 2\langle\boldsymbol{w}, \Theta_{\cdot,1}\rangle\langle\boldsymbol{w}, \Theta_{\cdot,2}\rangle}$$
$$= \frac{(\frac{13^2}{12^2} - 1)(\langle\boldsymbol{w}, \Theta_{\cdot,2}\rangle - (\langle\boldsymbol{w}, \Theta_{\cdot,1}\rangle + \langle\boldsymbol{w}, \Theta_{\cdot,2}\rangle)\theta_{i,2})^2}{\frac{13}{12}(\langle\boldsymbol{w}, \Theta_{\cdot,1}\rangle^2 + \langle\boldsymbol{w}, \Theta_{\cdot,2}\rangle^2) + 2\langle\boldsymbol{w}, \Theta_{\cdot,1}\rangle\langle\boldsymbol{w}, \Theta_{\cdot,2}\rangle}$$

Let $\frac{\langle\boldsymbol{w}, \Theta_{\cdot,1}\rangle}{\langle\boldsymbol{w}, \Theta_{\cdot,1}\rangle + \langle\boldsymbol{w}, \Theta_{\cdot,2}\rangle} = \alpha$. Further, Let $a, b$ be such that $a = \arg\max_i(\alpha - \theta_{i,1})^2$ and $b = \arg\max_i(1 - \alpha - \theta_{i,2})^2$. Then,

$$FRR(\alpha) \propto \frac{\max\{(\alpha - \theta_{a,1})^2, (1 - \alpha - \theta_{b,2})^2\}}{1 + \frac{\alpha^2 + (1-\alpha)^2}{12}}$$

To minimize the above expression w.r.t. $\alpha$, we compute the derivative of the above expression for each component of the max function

$$\frac{\partial\frac{(\alpha - \theta_{a,1})^2)}{1 + \frac{\alpha^2 + (1-\alpha)^2}{12}}}{\partial\alpha} \propto \frac{(\alpha - \theta_{a,1})(\alpha(2\theta_{a,1} - 1) - \theta_{a,1} + 13)}{(2\alpha - 2\theta_{a,1} + 13)^2}$$

and

$$\frac{\partial\frac{(1 - \alpha - \theta_{b,2})^2)}{1 + \frac{\alpha^2 + (1-\alpha)^2}{12}}}{\partial\alpha} \propto -\frac{(1 - \alpha + \theta_{b,2})(\alpha(2\theta_{b,2} - 1) - \theta_{b,1} - 12)}{(2\alpha - 2\theta_{a,1} + 13)^2}$$

For $\alpha \geq \frac{1+\theta_{a,1}-\theta_{b,2}}{2}$, the second term is greater and the derivative is positive. Similarly, for $\alpha \leq \frac{1+\theta_{a,1}-\theta_{b,2}}{2}$, the first term is greater and the derivative is positive. Hence, the minima is obtained at $\alpha = \frac{1+\theta_{a,1}-\theta_{b,2}}{2}$. Assuming $0.5 \leq \theta_{a,1}, \theta_{b,2} \leq 1$ or $0.0 \leq \theta_{a,1}, \theta_{b,2} \leq 0.5$.

Now, in order to compute $a, b$, we can look at the maximization problem again -

$$a = \arg\max_i (1 - (\theta_{i,1} + \theta_{b,2}))^2$$
$$b = \arg\max_i (1 - (\theta_{a,1} + \theta_{i,2}))^2$$

Since $\theta_{i,1}, \theta_{i,2}$ are bounded between $[0,1]$ the function to maximize is monotonically decreasing in $\theta_{\cdot,1}, \theta_{\cdot,2}$. Hence, $a = \arg\min_i \theta_{i,1}$ and $b = \arg\min_i \theta_{i,2}$. Conversely, $a = \arg\max_i \theta_{i,2}$, $b = \arg\max_i \theta_{i,1}$ and $\alpha = \frac{1+(\theta_{b,1}-\theta_{a,2})}{2}$. Hence,

$$\frac{\langle \boldsymbol{w}, \Theta_{\cdot,2} \rangle}{\langle \boldsymbol{w}, \Theta_{\cdot,1} \rangle} = \frac{1 - (\theta_{b,1} - \theta_{a,2})}{1 + (\theta_{b,1} - \theta_{a,2})}$$

Assuming that the maximum correlation of both the features is close, FRR will lead to a solution which gives roughly equal weights to both the features. $\qquad\square$

Note that for the case of feature replication, $\theta_{b,1} = 1$ and $\theta_{a,2} = 1$. Hence, $\frac{\langle \boldsymbol{w}, \Theta_{\cdot,2} \rangle}{\langle \boldsymbol{w}, \Theta_{\cdot,1} \rangle} = 1$.

# B    Justification of Feature Replication Hypothesis (FRH)

In a practical scenario where features are not disentangled, our hypothesis translates to the following:

**Conjecture:** Simpler features of the input are represented more in the feature space of neural networks, while complex (hard-to-learn) features are sparse.

**Assumptions:**

- We consider simple features such as background to be spurious, and complex features such as shape to be robust.
- We consider an overparameterized network that has the capacity to learn more features than what exist, resulting in feature repetition.

**Justification:** We justify the conjecture by showing that all other possibilities discussed below cannot be true.

1. **Assumption:** DNNs learn only Simple Features
   **Contradiction:** Prior works (Rosenfeld et al., 2022; Kirichenko et al., 2022b) show that features learned by ERM are diverse, and last layer training on target domain is good enough to obtain robustness to spurious features. This cannot be possible if the network has learned only spurious features.

2. **Assumption:** DNNs learn only Complex Features
   **Contradiction:** The dominance of Simple features in the learning of DNNs is shown by Shah et al. (2020). Moreover, the existence of texture-bias (Geirhos et al., 2018) and background-bias (Xiao et al., 2020) have been demonstrated in prior works, which show the dominance of Simple features.

3. **Assumption:** DNNs learn a uniform distribution of both Simple and Complex Features.
   **Contradiction:** SGD converges to an SVM solution due to its implicit bias (Soudry et al., 2018). From Claim-3.1 (1), in the presence of balanced features that are correlated with the label, SVM solution gives equal weight to all features to maximize margin. This contradicts the existence of Simplicity Bias (Shah et al., 2020).

4. **Assumption:** DNNs learn more Complex Features and less Simple Features.
   **Contradiction:** Since Complex features are indeed more robust and are better correlated with the labels, the classifier would rely more on these features. This contradicts the existence of Simplicity Bias (Shah et al., 2020).

Therefore, the only feasible option which supports the empirical observations in the prior works discussed above is that DNNs learn more Simple Features and Complex features are sparse, which justifies our conjecture.

## C  EMPIRICAL EVIDENCE FOR FEATURE REPLICATION HYPOTHESIS (FRH)

### C.1  SYNTHETIC DATASETS

We present empirical validation to support the Feature Replication Hypothesis (FRH) on several semi-real datasets and describe them in detail below:

1. **Coloured-MNIST-2** - In this dataset, we use images of digits superimposed on either of the two colours- red or green. The difference from Coloured-MNIST is that we consider only two colours for the background, rather than a range. We notice extreme simplicity bias in this case, with the network learning 32 colour features and 0 shape features.

2. **Coloured-MNIST-MultiDigit** - This is similar to the Coloured-MNIST dataset described in Section-3.1, with the exception that each of the classes is now composed of two digits. More specifically, the digits '1' and '7' and mapped to Class 0 and digits '5' and '8' are mapped to Class 1. We note that '1' and '5' are chosen from the original Coloured-MNIST dataset, while the second digit (e.g.'7') in each class is selected to be one that is similar to the first digit ('1') in the same class. This dataset is constructed specifically to show that the issue of Simplicity Bias and FRH exists even when there is higher variation in the shape feature, and is reported as ColouredMNIST-MultiDigit below. We see that while more shape features are learnt as compared to Coloured-MNIST, the network still relies more on colour to make its decisions.

3. **Digit-Coloured-MNIST**: This is similar to the Coloured-MNIST dataset described in Section-3.1, with the exception that the digit is coloured rather than the background. This dataset is constructed specifically to show that the issue of SB and FRH exist even when the region that is coloured, which is the extent to which simple features exist in the image is much lesser, and is reported as DigitColouredMNIST below. Although this dataset also demonstrates the presence of SB, we note that the average correlation of features with shape is higher when compared to the above datasets.

Table 4: Features learnt by an ERM trained model on synthetic datasets.

| Algorithm | Number | | Average correlation with input | | Correlation with output | | | |
|---|---|---|---|---|---|---|---|---|
| | Colour | Shape | Colour | Shape | Colour | Shape | ID Accuracy | OOD Accuracy |
| ColouredMNIST | 26 | 4 | 0.76 | 0.26 | 0.81 | 0.61 | 99.9 | 59.1 |
| TwoColouredMNIST | 32 | 0 | 0.90 | - | 0.82 | - | 99.9 | 49.5 |
| ColouredMNIST-MultDigit | 17 | 7 | 0.59 | 0.32 | 0.76 | 0.64 | 99.3 | 64.2 |
| DigitColouredMNIST | 26 | 3 | 0.76 | 0.36 | 0.79 | 0.45 | 99.9 | 62.1 |

### C.2  REAL WORLD EXAMPLE

We attempt to demonstrate feature replication in a model trained with ERM on the *Real* domain of OfficeHome. We train a ResNet-50 on this domain, and perform PCA on the features learnt by this network. The network learns 2048 features per example, and we compute the $2048 \times 2048$ sized covariance matrix of the features over samples from a test domain (Clipart). We then compute the eigenvalues of this matrix, and find that 500 principal components can explain about 97.5% of the variance, i.e. the matrix is extremely low rank, as shown in Fig.3. This points to the fact that a lot of the learnt features are linearly dependent and highly correlated with each other. This trend is similar to what we observed on ColouredMNIST, where a large number of features were highly correlated with the colour, and in turn with each other (Fig. 5 of the appendix).

We note that in all the additional datasets considered, simpler features are represented more in the network while complex (hard-to-learn) features are sparse. This empirically justifies our hypothesis in Section-3.

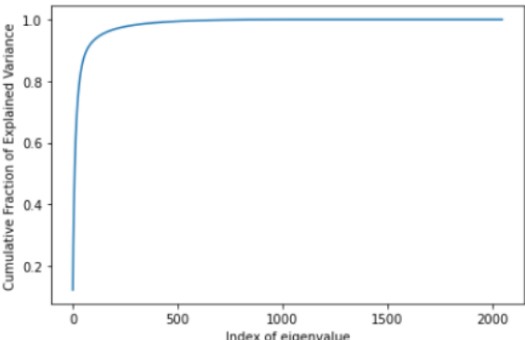

Figure 3: **Distribution of eigen values of covariance of learnt features**: A small fraction of principal components can explain most of the variance in features, indicating that features are highly correlated with each other.

# D  SYNTHETIC DATASETS

## D.1  COLOURED MNIST

In order to empirically demonstrate feature replication, we use a binarized version of the coloured MNIST dataset (Gulrajani & Lopez-Paz, 2020). To construct this dataset, we firstly assign two digits of the MNIST dataset, namely "1" and "5", to classes 0 and 1 respectively. For the in-domain training distribution, we associate colours in the range $R_0 = [(115, 0, 0) - (256, 141, 0))]$ (i.e. red) to label 0 (i.e. the digit "1") and the range $R_1 = [(0, 115, 0) - (141, 256, 0)]$ (i.e. green) to the label 1 (i.e. the digit "5"), where colors are represented in the RGB space. To summarize, while training the network, we super-impose images of "1" onto colours of range $R_0$, and images of "5" onto colours of range $R_1$. It is to be noted that the choice of colour ranges as defined above introduces an overlapping range between $[(115, 115, 0) - (141, 141, 0))]$ where images are associated with labels 0 and 1 with equal probability. This overlap reduces the correlation of colour features with labels, while shape features have a correlation of 1 with the labels. In Figure 4, we show examples of images from the train and test distributions of this dataset. In Figure 5, we pictorially depict the

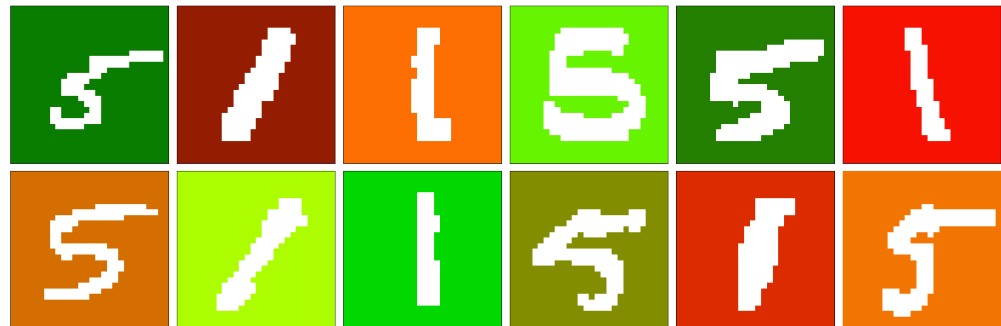

Figure 4: **Random images from the coloured MNIST dataset**: The top row shows examples from the train distribution, while the bottom row has images from the test distribution. Here, colour red corresponds to the digit 1 and green corresponds to the digit 5 in the train data, while this correlation is destroyed in the test data.

correlations between the 32 features learnt by the network. We can see a block structure emerging, indicating that there is a high amount of feature replication.

## D.2  TOY DATASET

In line with the theoretical formulation described in Section-3.3, we further empirically validate the brittleness of SVM models and the highlight the effectiveness of the proposed *Feature Reconstruc-*

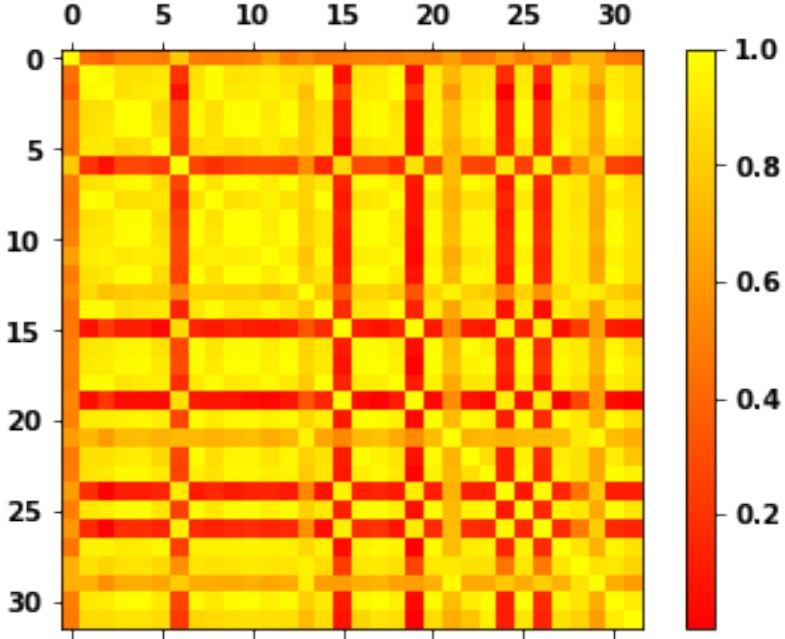

Figure 5: **Correlation of the features learnt on coloured MNIST**

*tion Regularizer* in the presence of replicated features. We consider a linearly separable toy distribution consisting of two factors of variation as shown in Figure 1. We define the means of the two classes at (1,1) and (-1,-1) and construct 500 data points in each class by adding noise sampled from Unif[-0.5,0.5] independently along each dimension to the respective means. We sample an Out-of-Distribution (OOD) test set from a Uniform distribution with means centered at (1,0) and (-1,0) respectively, and similar noise along each dimension as the train set. Therefore, while the train distribution can be classified by considering either the features aligned with the X-coordinate or the Y-coordinate, the test set performance crucially depends on the variation along X-coordinate alone. We consider feature replication along the y-axis, and hence construct this OOD dataset to verify the extent to which the other feature is considered for classification. To select the best hyperparameter for both SVM and FRR, we consider the presence of a validation set whose distribution is similar to the test distribution. As shown in Figure 1, we observe that the SVM model starts relying more on the replicated features alone in case of feature replication, compromising its performance on the OOD data. The proposed regularizer on the other hand, gives equal importance to both features even in the presence of feature replication, resulting in improved OOD generalization.

## E  ALGORITHM

Our training procedure is detailed in Alg 1.

## F  DETAILS ON THE OOD GENERALIZATION SETTING CONSIDERED

The problem of improving robustness to distribution shifts has been studied in several settings, where, in addition to labeled source domain data, varying levels of access to the target domain data is assumed. Some of the well-researched settings include - *Unsupervised Domain Adaptation*, with access to only unlabeled target domain data (Pan & Yang, 2009; Ganin et al., 2016), and *Domain Generalization*, where typically data from several source distributions is assumed to be available, and the target domain in unseen during training (Blanchard et al., 2011; Li et al., 2018a; Gulrajani & Lopez-Paz, 2020)). In the latter case, it is assumed that all training data samples are annotated with domain labels as well, so that training algorithms can explicitly impose invariance to attributes

---

**Algorithm 1:** Our training algorithm

---

**Data:** Training data $\mathcal{D}_S = \{(x_i, y_i) : i \in [n]\}$, model $(\theta, W)$, feature reconstruction model $\phi$,
$\qquad \lambda_{FRR}, \lambda_{FT}$

1 $\theta_{\text{std}}, W_{\text{std}} \leftarrow \text{Adam}\left(\min_{\theta, W} \sum_i \mathcal{L}_{\text{std}}(\theta, W, (x_i, y_i))\right).$
$\qquad\qquad$ /* Standard training of model parameters $\theta$ and $W$. */

2 Freeze $\theta$ to be $\theta_{std}$
$\qquad\qquad\qquad$ /* Initializing model for training with FRR. */

3 $W_{FRR}, \phi_{FRR} \leftarrow \text{Adam}\left(\min_{W,\phi} \sum_i \mathcal{L}_{\text{std}}(\theta_{std}, W, (x_i, y_i)) + \lambda_L \mathcal{L}_{FRR}(x_i, \theta_{std}, W, \phi)\right).$
$\qquad\qquad$ /* FRR-L: Training $W, \phi$ with FRR defined in eqn. 4 */

4 $\theta_{\text{FLFT}} \leftarrow \text{Adam}\left(\min_\theta \sum_i \mathcal{L}_{\text{std}}(\theta, W_{\text{FRR}}, (x_i, y_i)) + \lambda_{FLFT} \mathcal{L}_{FRR}(x_i, \theta, W_{FRR}, \phi_{FRR})\right).$
$\qquad\quad$ /* FRR-FLFT: Finetuning $\theta$ with FRR according to eqn. 5 */

**Result:** Trained model $(\theta_{\text{FLFT}}, W_{\text{FRR}})$.

---

that cause a distribution shift in input data without change in their label distribution (Muandet et al., 2013; Ganin et al., 2016; Li et al., 2018b; Arjovsky et al., 2019; Shi et al., 2021).

A more challenging case is when the training data belongs to several distributions that may not even be sufficiently discernable to have explicit domain annotations, or may contain multidimensional distribution shifts, such as weather, time of the day and geographical location, that cannot be easily annotated or clustered. We investigate this crucial setting which has been relatively less researched, and refer to it as *Aggregated Domain Generalization*, as introduced by Thomas et al. (2021). We note that this setting is different from the case of training on data from a single domain such as ImageNet, and evaluating on distribution shifts (Hendrycks & Dietterich, 2019), due to the availability of an *aggregate* of source domains during training, which can enable the effective use of in-domain validation set for hyperparameter selection.

While there have been several approaches to improve the performance of models in the setting of Domain Generalization, Gulrajani & Lopez-Paz (2020) show that when evaluated fairly, that is, without assuming access to the test domain data even for selecting the best set of hyperparameters, none of the approaches perform consistently better than standard training using *Empirical Risk Minimization (ERM)*. Furthermore, we consider the setting of *Aggregated Domain Generalization*, which is more challenging due to the absence of domain labels during both training and validation.

## G    EXPERIMENTAL DETAILS ON DOMAINBED

We test our approach on the DomainBed benchmark (Gulrajani & Lopez-Paz, 2020) comprising of five different datasets, each of which have $k$ domains. For each dataset, we train a model on $k-1$ domains, and test it on the left out domain. The average out-of-domain performance across the $k$ held-out domains is then reported. In this section we describe the hyper-parameter selection strategy and the ranges considered for our approach. In line with the DomainBed testbench, we use ImageNet pretrained ResNet-50 models for all algorithms. We use random search to select hyperparameters for our algorithm, and use the suggested hyperparameters for the other baselines. We train for 3000 (5000 for DomainNet) steps in the FRR-L phase, and 5000 (10000 for DomainNet) steps in the FRR-FLFT phase. The batch size is fixed to 32, and SWAD hyper-parameters are the same as those used by Cha et al. (2021). We use the in-domain accuracy protocol from Gulrajani & Lopez-Paz (2020) to select hyper-parameters for each domain of each dataset, and search over 8 random combinations of hyper-parameters for each. The range of the hyperparameters is shown in Table 5. Note that we experiment with two implementations of $\ell_\infty$ norm: $\ell_{1,\infty}$, where we first compute the $\ell_\infty$ of feature reconstruction for each example in a batch and then average it across the batch, and $\ell_{\infty,1}$ where we compute the average $\ell_1$ reconstruction norm of each feature across the batch, and then apply $\ell_\infty$ norm on this $m$ dimensional vector. All our experiments were done on single V100 GPUs.

Table 5: **Ranges of hyperparameters considered for DomainBed**

| Hparam | Range |
|---|---|
| Learning Rate | $\text{loguniform}(10^{-5}, 10^{-1})$ |
| $\lambda_{\text{FRR}}$ | $\text{loguniform}(10^{-6}, 10^{0})$ |
| $\lambda_{\text{FT}}$ | $\text{loguniform}(10^{-6}, 10^{0})$ |
| Norm | $\{\ell_1, \ell_{1,\infty}, \ell_{\infty,1}\}$ |

## H ABLATIONS ON DOMAINBED

**Comparing the choices for $\phi$** : In Table 6, we experiment with various architectures for the decoder $\phi$ when computing FRR according to equation 1. We consider using a two layer neural network as the decoder $\phi$ (FRR-LDeeper), and also consider setting $\phi = W^T$ (FRR-LShared), i.e. explicitly tying the weights of the decoder and the classifier layer. Overall, both these variants are worse than the default single layer, free parameterization of $\phi$. We believe that this happens because the latter approach enforces a much stricter constraint on $W$, leading to poorer in-domain accuracy, while the former approach enforces a weaker constraint, potentially enabling reconstruction of more complex features from a smaller amount of information about them in the logits. Both these have a detrimental effect on the overall performance of the model.

Table 6: Effect of different design choices on OOD accuracy:the rows shows different architecture choices for $\phi$

| Algorithm | PACS | OfficeHome | TerraIncognita | Avg. |
|---|---|---|---|---|
| ERM | $85.5 _{\pm 0.1}$ | $66.5 _{\pm 0.2}$ | $46.1 _{\pm 0.6}$ | 65.3 |
| ERM+FRR-LShared | $85.2 _{\pm 0.5}$ | $68.2 _{\pm 0.1}$ | $49.4 _{\pm 0.5}$ | 67.6 |
| ERM+FRR-LDeeper | $84.6 _{\pm 0.7}$ | $65.6 _{\pm 0.2}$ | $52.5 _{\pm 0.5}$ | 67.6 |

**Sensitivity Analysis** : We vary $\lambda_{FRR}$ and plot out the OOD performance in Fig 6. We find that the performance is stable for a wide range of the hyper-parameter on most domains.

## I PSEUDO-CODE FOR FRR

Below we provide the python code for FRR-L in the DomainBed framework.

```python
class ERMWithFRR_L(Algorithm):

    def __init__(self, input_shape, num_classes, num_domains, hparams):
        super(ERMWithFRR_L, self).__init__(
            input_shape, num_classes, num_domains, hparams
        )
        self.featurizer = networks.Featurizer(input_shape, self.hparams)
        self.classifier = networks.Classifier(
            self.featurizer.n_outputs,
            num_classes,
            self.hparams['nonlinear_classifier'],
        )
        for params in self.featurizer.parameters():
            params.requires_grad = False
        self.classifier_inv = networks.Classifier(
            num_classes,
            self.featurizer.n_outputs,
            self.hparams['nonlinear_classifier'],
```

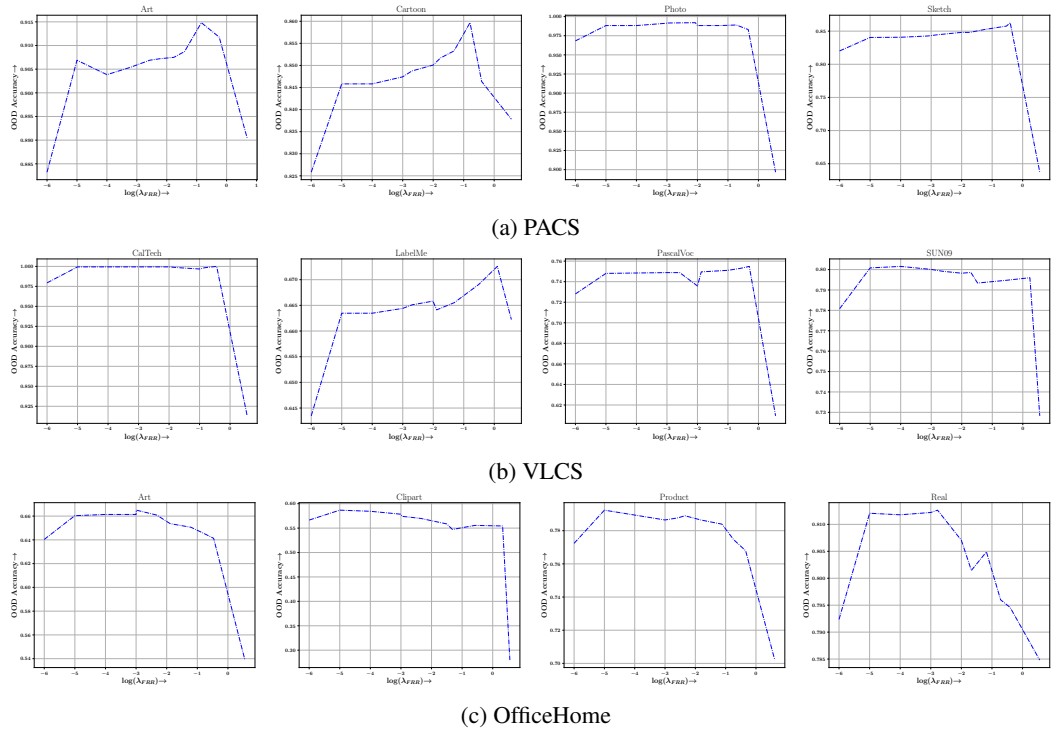

(a) PACS

(b) VLCS

(c) OfficeHome

Figure 6: Variation of OOD accuracy with varying $\lambda_{FRR}$

```
        bias=False
        if 'exact_reconstruction' in self.hparams
        and self.hparams['exact_reconstruction']
        else True,
    )
    self.network = nn.Sequential(self.featurizer, self.classifier)

    self.optimizer = torch.optim.Adam(
        (
            list(self.network.parameters())
            + list(self.classifier_inv.parameters())
        ),
        lr=self.hparams['lr'],
        weight_decay=self.hparams['weight_decay'],
    )
    self.reconstruction_wt = self.hparams['reconstruction_wt']
    self.norm = float(self.hparams['norm'])

def update(self, minibatches, unlabeled=None):
    all_x = torch.cat([x for x, y in minibatches])
    all_y = torch.cat([y for x, y in minibatches])
    pred, rec, feat = self.get_feats_and_rec(all_x)
    loss = F.cross_entropy(pred, all_y)
    reconstruction_loss = (
        torch.sum(torch.max(torch.abs(feat - rec), dim=1)[0])
        / all_x.shape[0]
    )
    loss = loss + self.reconstruction_wt * reconstruction_loss
    self.optimizer.zero_grad()
    loss.backward()
```

```
    self.optimizer.step()

    return {
        'loss': loss.item(),
        'reconstruction_loss': reconstruction_loss.item(),
    }

def predict(self, x):
    return self.network(x)

def get_feats_and_rec(self, x):
    feats = self.network[0](x)
    pred = self.network[1](feats)
    rec = self.classifier_inv(pred)
    return pred, rec, feats
```

## J   DOMAIN WISE ACCURACIES

In this section, we show detailed results of Table 3 in the main text. The numbers for the baselines are taken from Gulrajani & Lopez-Paz (2020), Cha et al. (2021) and Arpit et al. (2021), while the results for MIRO (Cha et al., 2022) were reproduced using their code-base.

Table 7: **Out-of-domain accuracies (%) on** PACS.

| Algorithm | A | C | P | S | Avg |
|---|---|---|---|---|---|
| CDANN | $84.6 \pm 1.8$ | $75.5 \pm 0.9$ | $96.8 \pm 0.3$ | $73.5 \pm 0.6$ | 82.6 |
| MASF | 82.9 | 80.5 | 95.0 | 72.3 | 82.7 |
| DMG | 82.6 | 78.1 | 94.5 | 78.3 | 83.4 |
| IRM | $84.8 \pm 1.3$ | $76.4 \pm 1.1$ | $96.7 \pm 0.6$ | $76.1 \pm 1.0$ | 83.5 |
| MetaReg | 87.2 | 79.2 | 97.6 | 70.3 | 83.6 |
| DANN | $86.4 \pm 0.8$ | $77.4 \pm 0.8$ | $97.3 \pm 0.4$ | $73.5 \pm 2.3$ | 83.7 |
| GroupDRO | $83.5 \pm 0.9$ | $79.1 \pm 0.6$ | $96.7 \pm 0.3$ | $78.3 \pm 2.0$ | 84.4 |
| MTL | $87.5 \pm 0.8$ | $77.1 \pm 0.5$ | $96.4 \pm 0.8$ | $77.3 \pm 1.8$ | 84.6 |
| I-Mixup | $86.1 \pm 0.5$ | $78.9 \pm 0.8$ | $97.6 \pm 0.1$ | $75.8 \pm 1.8$ | 84.6 |
| MMD | $86.1 \pm 1.4$ | $79.4 \pm 0.9$ | $96.6 \pm 0.2$ | $76.5 \pm 0.5$ | 84.7 |
| VREx | $86.0 \pm 1.6$ | $79.1 \pm 0.6$ | $96.9 \pm 0.5$ | $77.7 \pm 1.7$ | 84.9 |
| MLDG | $85.5 \pm 1.4$ | $80.1 \pm 1.7$ | $97.4 \pm 0.3$ | $76.6 \pm 1.1$ | 84.9 |
| ARM | $86.8 \pm 0.6$ | $76.8 \pm 0.5$ | $97.4 \pm 0.3$ | $79.3 \pm 1.2$ | 85.1 |
| RSC | $85.4 \pm 0.8$ | $79.7 \pm 1.8$ | $97.6 \pm 0.3$ | $78.2 \pm 1.2$ | 85.2 |
| Mixstyle | $86.8 \pm 0.5$ | $79.0 \pm 1.4$ | $96.6 \pm 0.1$ | $78.5 \pm 2.3$ | 85.2 |
| ER | 87.5 | 79.3 | 98.3 | 76.3 | 85.3 |
| pAdaIN | 85.8 | 81.1 | 97.2 | 77.4 | 85.4 |
| ERM | $84.7 \pm 0.4$ | $80.8 \pm 0.6$ | $97.2 \pm 0.3$ | $79.3 \pm 1.0$ | 85.5 |
| EISNet | 86.6 | 81.5 | 97.1 | 78.1 | 85.8 |
| CORAL | $88.3 \pm 0.2$ | $80.0 \pm 0.5$ | $97.5 \pm 0.3$ | $78.8 \pm 1.3$ | 86.2 |
| SagNet | $87.4 \pm 1.0$ | $80.7 \pm 0.6$ | $97.1 \pm 0.1$ | $80.0 \pm 0.4$ | 86.3 |
| DSON | 87.0 | 80.6 | 96.0 | 82.9 | 86.6 |
| SMA | $89.1 \pm 0.1$ | $82.6 \pm 0.2$ | $97.6 \pm 0.0$ | $80.5 \pm 0.9$ | 87.5 |
| MIRO | 87.5 | 79.0 | 98.3 | 76.2 | 85.3 |
| SWAD | $89.3 \pm 0.2$ | $83.4 \pm 0.6$ | $97.3 \pm 0.3$ | $82.5 \pm 0.5$ | 88.1 |
| SWAD+FRR | $89.9 \pm 0.2$ | $83.9 \pm 0.7$ | $98.2 \pm 0.3$ | $84.8 \pm 0.4$ | 89.2 |

Table 8: **Out-of-domain accuracies (%) on** `VLCS`.

| Algorithm | C | L | S | V | Avg |
|---|---|---|---|---|---|
| GroupDRO | 97.3 ± 0.3 | 63.4 ± 0.9 | 69.5 ± 0.8 | 76.7 ± 0.7 | 76.7 |
| RSC | 97.9 ± 0.1 | 62.5 ± 0.7 | 72.3 ± 1.2 | 75.6 ± 0.8 | 77.1 |
| MLDG | 97.4 ± 0.2 | 65.2 ± 0.7 | 71.0 ± 1.4 | 75.3 ± 1.0 | 77.2 |
| MTL | 97.8 ± 0.4 | 64.3 ± 0.3 | 71.5 ± 0.7 | 75.3 ± 1.7 | 77.2 |
| I-Mixup | 98.3 ± 0.6 | 64.8 ± 1.0 | 72.1 ± 0.5 | 74.3 ± 0.8 | 77.4 |
| ERM | 97.7 ± 0.4 | 64.3 ± 0.9 | 73.4 ± 0.5 | 74.6 ± 1.3 | 77.5 |
| MMD | 97.7 ± 0.1 | 64.0 ± 1.1 | 72.8 ± 0.2 | 75.3 ± 3.3 | 77.5 |
| CDANN | 97.1 ± 0.3 | 65.1 ± 1.2 | 70.7 ± 0.8 | 77.1 ± 1.5 | 77.5 |
| ARM | 98.7 ± 0.2 | 63.6 ± 0.7 | 71.3 ± 1.2 | 76.7 ± 0.6 | 77.6 |
| SagNet | 97.9 ± 0.4 | 64.5 ± 0.5 | 71.4 ± 1.3 | 77.5 ± 0.5 | 77.8 |
| Mixstyle | 98.6 ± 0.3 | 64.5 ± 1.1 | 72.6 ± 0.5 | 75.7 ± 1.7 | 77.9 |
| VREx | 98.4 ± 0.3 | 64.4 ± 1.4 | 74.1 ± 0.4 | 76.2 ± 1.3 | 78.3 |
| IRM | 98.6 ± 0.1 | 64.9 ± 0.9 | 73.4 ± 0.6 | 77.3 ± 0.9 | 78.6 |
| DANN | 99.0 ± 0.3 | 65.1 ± 1.4 | 73.1 ± 0.3 | 77.2 ± 0.6 | 78.6 |
| CORAL | 98.3 ± 0.1 | 66.1 ± 1.2 | 73.4 ± 0.3 | 77.5 ± 1.2 | 78.8 |
| SMA | 99.0 ± 0.2 | 63.0 ± 0.2 | 74.5 ± 0.3 | 76.4 ± 1.1 | 78.2 |
| MIRO | 99.3 | 65.2 | 74.9 | 76.0 | 78.9 |
| SWAD | 98.8 ± 0.1 | 63.3 ± 0.3 | 75.3 ± 0.5 | 79.2 ± 0.6 | 79.1 |
| SWAD+FRR | 98.9 ± 0.4 | 66.3 ± 0.2 | 75.9 ± 0.6 | 79.0 ± 0.2 | 80.0 |

Table 9: **Out-of-domain accuracies (%) on** `OfficeHome`.

| Algorithm | A | C | P | R | Avg |
|---|---|---|---|---|---|
| Mixstyle | 51.1 ± 0.3 | 53.2 ± 0.4 | 68.2 ± 0.7 | 69.2 ± 0.6 | 60.4 |
| IRM | 58.9 ± 2.3 | 52.2 ± 1.6 | 72.1 ± 2.9 | 74.0 ± 2.5 | 64.3 |
| ARM | 58.9 ± 0.8 | 51.0 ± 0.5 | 74.1 ± 0.1 | 75.2 ± 0.3 | 64.8 |
| RSC | 60.7 ± 1.4 | 51.4 ± 0.3 | 74.8 ± 1.1 | 75.1 ± 1.3 | 65.5 |
| CDANN | 61.5 ± 1.4 | 50.4 ± 2.4 | 74.4 ± 0.9 | 76.6 ± 0.8 | 65.7 |
| DANN | 59.9 ± 1.3 | 53.0 ± 0.3 | 73.6 ± 0.7 | 76.9 ± 0.5 | 65.9 |
| GroupDRO | 60.4 ± 0.7 | 52.7 ± 1.0 | 75.0 ± 0.7 | 76.0 ± 0.7 | 66.0 |
| MMD | 60.4 ± 0.2 | 53.3 ± 0.3 | 74.3 ± 0.1 | 77.4 ± 0.6 | 66.4 |
| MTL | 61.5 ± 0.7 | 52.4 ± 0.6 | 74.9 ± 0.4 | 76.8 ± 0.4 | 66.4 |
| VREx | 60.7 ± 0.9 | 53.0 ± 0.9 | 75.3 ± 0.1 | 76.6 ± 0.5 | 66.4 |
| ERM | 61.3 ± 0.7 | 52.4 ± 0.3 | 75.8 ± 0.1 | 76.6 ± 0.3 | 66.5 |
| MLDG | 61.5 ± 0.9 | 53.2 ± 0.6 | 75.0 ± 1.2 | 77.5 ± 0.4 | 66.8 |
| I-Mixup | 62.4 ± 0.8 | 54.8 ± 0.6 | 76.9 ± 0.3 | 78.3 ± 0.2 | 68.1 |
| SagNet | 63.4 ± 0.2 | 54.8 ± 0.4 | 75.8 ± 0.4 | 78.3 ± 0.3 | 68.1 |
| CORAL | 65.3 ± 0.4 | 54.4 ± 0.5 | 76.5 ± 0.1 | 78.4 ± 0.5 | 68.7 |
| SMA | 66.7 ± 0.5 | 57.1 ± 0.1 | 78.6 ± 0.1 | 80.0 ± 0 | 70.6 |
| MIRO | 66.0 | 54.5 | 78.9 | 81.7 | 70.3 |
| SWAD | 66.1 ± 0.4 | 57.7 ± 0.4 | 78.4 ± 0.1 | 80.2 ± 0.2 | 70.6 |
| SWAD+FRR | 65.2 ± 0.2 | 57.7 ± 0.5 | 78.2 ± 0.2 | 80.2 ± 0.1 | 70.3 |

Table 10: **Out-of-domain accuracies (%) on** `TerraIncognita`.

| Algorithm | L100 | L38 | L43 | L46 | Avg |
|---|---|---|---|---|---|
| MMD | 41.9 ± 3.0 | 34.8 ± 1.0 | 57.0 ± 1.9 | 35.2 ± 1.8 | 42.2 |
| GroupDRO | 41.2 ± 0.7 | 38.6 ± 2.1 | 56.7 ± 0.9 | 36.4 ± 2.1 | 43.2 |
| Mixstyle | 54.3 ± 1.1 | 34.1 ± 1.1 | 55.9 ± 1.1 | 31.7 ± 2.1 | 44.0 |
| ARM | 49.3 ± 0.7 | 38.3 ± 2.4 | 55.8 ± 0.8 | 38.7 ± 1.3 | 45.5 |
| MTL | 49.3 ± 1.2 | 39.6 ± 6.3 | 55.6 ± 1.1 | 37.8 ± 0.8 | 45.6 |
| CDANN | 47.0 ± 1.9 | 41.3 ± 4.8 | 54.9 ± 1.7 | 39.8 ± 2.3 | 45.8 |
| ERM | 49.8 ± 4.4 | 42.1 ± 1.4 | 56.9 ± 1.8 | 35.7 ± 3.9 | 46.1 |
| VREx | 48.2 ± 4.3 | 41.7 ± 1.3 | 56.8 ± 0.8 | 38.7 ± 3.1 | 46.4 |
| RSC | 50.2 ± 2.2 | 39.2 ± 1.4 | 56.3 ± 1.4 | 40.8 ± 0.6 | 46.6 |
| DANN | 51.1 ± 3.5 | 40.6 ± 0.6 | 57.4 ± 0.5 | 37.7 ± 1.8 | 46.7 |
| IRM | 54.6 ± 1.3 | 39.8 ± 1.9 | 56.2 ± 1.8 | 39.6 ± 0.8 | 47.6 |
| CORAL | 51.6 ± 2.4 | 42.2 ± 1.0 | 57.0 ± 1.0 | 39.8 ± 2.9 | 47.7 |
| MLDG | 54.2 ± 3.0 | 44.3 ± 1.1 | 55.6 ± 0.3 | 36.9 ± 2.2 | 47.8 |
| I-Mixup | 59.6 ± 2.0 | 42.2 ± 1.4 | 55.9 ± 0.8 | 33.9 ± 1.4 | 47.9 |
| SagNet | 53.0 ± 2.9 | 43.0 ± 2.5 | 57.9 ± 0.6 | 40.4 ± 1.3 | 48.6 |
| SMA | 54.9 ± 0.4 | 45.5 ± 0.6 | 60.1 ± 1.5 | 40.5 ± 0.4 | 50.3 |
| MIRO | 59.6 | 41.1 | 60.2 | 40.4 | 50.3 |
| SWAD | 55.4 ± 0.0 | 44.9 ± 1.1 | 59.7 ± 0.4 | 39.9 ± 0.2 | 50.0 |
| SWAD+FRR | 60.13 ± 1.05 | 47.89 ± 1.71 | 60.76 ± 0.42 | 42.34 ± 1.35 | 53.2 |

Table 11: **Out-of-domain accuracies (%) on** `DomainNet`.

| Algorithm | clip | info | paint | quick | real | sketch | Avg |
|---|---|---|---|---|---|---|---|
| MMD | 32.1 ± 13.3 | 11.0 ± 4.6 | 26.8 ± 11.3 | 8.7 ± 2.1 | 32.7 ± 13.8 | 28.9 ± 11.9 | 23.4 |
| GroupDRO | 47.2 ± 0.5 | 17.5 ± 0.4 | 33.8 ± 0.5 | 9.3 ± 0.3 | 51.6 ± 0.4 | 40.1 ± 0.6 | 33.3 |
| VREx | 47.3 ± 3.5 | 16.0 ± 1.5 | 35.8 ± 4.6 | 10.9 ± 0.3 | 49.6 ± 4.9 | 42.0 ± 3.0 | 33.6 |
| IRM | 48.5 ± 2.8 | 15.0 ± 1.5 | 38.3 ± 4.3 | 10.9 ± 0.5 | 48.2 ± 5.2 | 42.3 ± 3.1 | 33.9 |
| Mixstyle | 51.9 ± 0.4 | 13.3 ± 0.2 | 37.0 ± 0.5 | 12.3 ± 0.1 | 46.1 ± 0.3 | 43.4 ± 0.4 | 34.0 |
| ARM | 49.7 ± 0.3 | 16.3 ± 0.5 | 40.9 ± 1.1 | 9.4 ± 0.1 | 53.4 ± 0.4 | 43.5 ± 0.4 | 35.5 |
| CDANN | 54.6 ± 0.4 | 17.3 ± 0.1 | 43.7 ± 0.9 | 12.1 ± 0.7 | 56.2 ± 0.4 | 45.9 ± 0.5 | 38.3 |
| DANN | 53.1 ± 0.2 | 18.3 ± 0.1 | 44.2 ± 0.7 | 11.8 ± 0.1 | 55.5 ± 0.4 | 46.8 ± 0.6 | 38.3 |
| RSC | 55.0 ± 1.2 | 18.3 ± 0.5 | 44.4 ± 0.6 | 12.2 ± 0.2 | 55.7 ± 0.7 | 47.8 ± 0.9 | 38.9 |
| I-Mixup | 55.7 ± 0.3 | 18.5 ± 0.5 | 44.3 ± 0.5 | 12.5 ± 0.4 | 55.8 ± 0.3 | 48.2 ± 0.5 | 39.2 |
| SagNet | 57.7 ± 0.3 | 19.0 ± 0.2 | 45.3 ± 0.3 | 12.7 ± 0.5 | 58.1 ± 0.5 | 48.8 ± 0.2 | 40.3 |
| MTL | 57.9 ± 0.5 | 18.5 ± 0.4 | 46.0 ± 0.1 | 12.5 ± 0.1 | 59.5 ± 0.3 | 49.2 ± 0.1 | 40.6 |
| ERM | 58.1 ± 0.3 | 18.8 ± 0.3 | 46.7 ± 0.3 | 12.2 ± 0.4 | 59.6 ± 0.1 | 49.8 ± 0.4 | 40.9 |
| MLDG | 59.1 ± 0.2 | 19.1 ± 0.3 | 45.8 ± 0.7 | 13.4 ± 0.3 | 59.6 ± 0.2 | 50.2 ± 0.4 | 41.2 |
| CORAL | 59.2 ± 0.1 | 19.7 ± 0.2 | 46.6 ± 0.3 | 13.4 ± 0.4 | 59.8 ± 0.2 | 50.1 ± 0.6 | 41.5 |
| MetaReg | 59.8 | 25.6 | 50.2 | 11.5 | 64.6 | 50.1 | 43.6 |
| DMG | 65.2 | 22.2 | 50.0 | 15.7 | 59.6 | 49.0 | 43.6 |
| SMA | 64.4 ± 0.3 | 22.4 ± 0.2 | 53.4 ± 0.3 | 15.4 ± 0.1 | 64.7 ± 0.2 | 55.5 ± 0.1 | 46.0 |
| MIRO | 61.9 | 20.9 | 50.3 | 13.0 | 65.2 | 52.7 | 44.2 |
| SWAD | 66.0 ± 0.1 | 22.4 0.3 | 53.5 ± 0.1 | 16.1 ± 0.2 | 65.8 ± 0.4 | 55.5 ± 0.3 | 46.5 |
| SWAD+FRR | 65.9 ± 0.1 | 22.3 ± 0.0 | 52.8 ± 0.1 | 14.8 ± 0.3 | 66.2 ± 0.1 | 55.0 ± 0.1 | 46.2 |

