# OpenReview forum: "Feature Reconstruction From Outputs Can Mitigate Simplicity Bias in Neural Networks"
_ICLR.cc/2023/Conference — ICLR 2023 poster_

### Official Review · Reviewer_PdgB · 2022-10-22

**Confidence:** 4
**Correctness:** 3
**Technical Novelty And Significance:** 3
**Empirical Novelty And Significance:** 3
**Recommendation:** 5

**Clarity, Quality, Novelty And Reproducibility:**

The clarity of the paper can be improved:
- In Figure 1, (c) and (d) have exactly the same title, but with different contents
- The author mentioned that, in section 3.1, that the number of learnt color feature, and "90% correlation with the color in the input". The reviewer read through paper and did not find how does the author calculate the correlation between a feature and the input.
- Following the previous point, how does the number of learnt features are counted? Does the author count each dimension of the feature space as a feature?
- In proposition 3.2 the loss is a infinity norm minimization problem, while in the proof, somehow the analytic solution for least square is provided. This is very confusing, please provide clarification about this.



**Strength And Weaknesses:**

**Strength**:
- The paper spot an interesting observation that even though the complex features are learned, the amount of it is significantly less than the simple ones.
- A feature reconstruction regularization is proposed to enhance robustness.

**Weakness**:
- Lack of comprehensive analysis on the observed phenomenon.
- The writing of the paper is not clear.
- More experiments need to be done such as on larger datasets and comparing with other SoTA methods.

**Summary Of The Paper:**

Inspired by the previous observation that the network is able to learn complex features even when it is biasing to simple features. This paper propose to further investigate how much the complex features are learned compared with the simple features. They find that the simple features are replicated multiple times over complicated ones. Based on this observation, they propose a novel regularization term and verify the effectiveness with some experiments.

**Summary Of The Review:**

Overall, the authors observe a very interesting phenomenon, based on the observation, a regularization term is proposed. But the reviewer feels more comprehensive analysis can be done and included, and the quality of the current version of the paper is not yet ready for publication in ICLR.

---

> ### Author Response · Authors · 2022-11-11
> **Response to Reviewer PdgB (1)**
>
> **A note to all reviewers:**
>
> We sincerely thank the reviewers for their time and valuable feedback on our work. At the outset, it is encouraging that all four reviewers find our observation on *Feature Replication Hypothesis (FRH)* interesting and novel, and some reviewers find our FRR regularizer easy to understand and implement, experiments promising, and writing to be clear with a logical flow.
>
> We have identified two common concerns in all reviews -
>
> - **Justification of FRH**: While we intended to show this as an assumption, and hence term this as a *Hypothesis*, we understand that since this is a key component and contribution of the paper, it needs better justification.
>
> - **Coloured MNIST**: We note that a second common concern was the explanation of the Colored MNIST experiments.
>
> We justify these major concerns and the remaining points raised by each reviewer in our rebuttal.
>
> We believe the insightful feedback from all the reviewers has indeed helped us identify and strengthen the weak links in our work. We hope to have a fruitful discussion through the discussion period, which can be very helpful in making this work a valuable contribution to the research community. We have also updated the draft with our responses in appropriate sections. The updates in the draft have been done in blue for better clarity.
>
> <Response continues in the next post, Response to Reviewer PdgB (2) >

---

> > ### Author Response · Authors · 2022-11-11
> > **Response to Reviewer PdgB (2)**
> >
> > < Continued from Response to Reviewer PdgB (1) >
> >
> > **Comprehensive analysis on the observed phenomenon**
> >    We present a comprehensive analysis of the observed phenomenon of Feature Replication Hypothesis (FRH) below:
> >
> >  **Formal Proof of FRH:** We do not present a theoretical proof of FRH, and hence refer to it as a *Hypothesis*. However, for better understanding, we state our intuition more formally below -
> >
> > In a practical scenario where features are not disentangled, our hypothesis translates to the following:
> >
> >   **Conjecture:** Simpler features of the input are represented more in the feature space of neural networks, while complex (hard-to-learn) features are sparse.
> >
> >   **Assumptions:**
> >   -  We consider simple features such as background to be spurious, and complex features such as shape to be robust.
> >   -  We consider an overparameterized network that has the capacity to learn more features than what exist, resulting in feature repetition.
> >
> >   **Justification:** We justify the conjecture by showing that all other possibilities discussed below cannot be true.
> >
> >   1. Assumption: DNNs learn only Simple Features
> >
> >      Contradiction: Prior works [2,3] show that features learned by ERM are diverse, and last layer training on target domain is good enough to obtain robustness to spurious features. This cannot be possible if the network has learned only spurious features.
> >
> >   2. Assumption: DNNs learn only Complex Features
> >
> >      Contradiction: The dominance of Simple features in the learning of DNNs is shown by Shah et al. [5]. Moreover, the existence of texture-bias [6] and background-bias [7] have been demonstrated in prior works, which show the dominance of Simple features.
> >   3. Assumption: DNNs learn a uniform distribution of both Simple and Complex Features.
> >
> >       Contradiction: SGD converges to a max-margin solution due to its implicit bias. From Claim-3.1 (1), in the presence of balanced features that are correlated with the label, max-margin solution gives equal weight to all features. This contradicts the existence of Simplicity Bias [5].
> >
> >   4. Assumption: DNNs learn more Complex Features and less Simple Features.
> >
> >      Contradiction: Since Complex features are indeed more robust and are better correlated with the labels, the classifier would rely more on these features. This contradicts the existence of Simplicity Bias [5].
> >
> >   Therefore, the only feasible option which supports the empirical observations in prior works [1,2,3,4,5,6,7] is that DNNs learn more Simple Features and Complex features are sparse, which justifies our conjecture.
> >
> > - **Empirical Justification of Feature Replication Hypothesis (FRH) on Real Datasets:** We justified FRH using only the toy dataset because it is a synthetic setting where we are aware of the good and spurious features, making it very easy to compute correlations. The same cannot be done in a real dataset since there are several good and spurious features. Moreover, we are not explicitly aware of the spurious features. However, we understand that demonstrating FRH on a real dataset is very important to strengthen our hypothesis. We thus attempt to show that Feature Replication is seen even in Real datasets using a different experimental setting as described below.
> >
> >   We attempt to demonstrate feature replication in a model trained with ERM on the *Real* domain of OfficeHome. We train a ResNet-50 on this domain, and perform PCA on the features learnt by this network. The network learns 2048 features per example, and we compute the $2048 \times 2048$ sized covariance matrix of the features over samples from a test domain (Clipart). We then compute the eigenvalues of this matrix, and find that 500 principal components can explain about 97.5\% of the variance, i.e. the matrix is extremely low rank, as shown in Fig.3 of the Appendix. This points to the fact that a lot of the learnt features are linearly dependent and highly correlated with each other. This trend is similar to what we observed on ColouredMNIST, where a large number of features were highly correlated with the colour, and in turn with each other.
> >
> > We demonstrate our hypothesis on other synthetic datasets in Appendix-C of the updated submission.
> >
> > [1] Soudry et al., The implicit bias of gradient descent on separable data, JMLR 2018
> >
> > [2] Rosenfeld et al., Domain-adjusted regression or: Erm may already learn features sufficient for out-of-distribution generalization, 2022
> >
> > [3] Kirichenko et al., Last layer re-training is sufficient for robustness to spurious correlations, 2022
> >
> > [4] Gulrajani et al., In search of lost domain generalization, 2020
> >
> > [5] Shah et al., The pitfalls of simplicity bias in neural networks, NeurIPS 2020
> >
> > [6] Geirhos et al., ImageNet-trained CNNs are biased towards texture; increasing shape bias improves accuracy and robustness, ICLR 2019
> >
> > [7] Xiao et al., Noise or Signal: The Role of Image Backgrounds in Object Recognition, ICLR 2021
> >
> > <Continued in Response to Reviewer PdgB (3) >

---

> > > ### Author Response · Authors · 2022-11-11
> > > **Response to Reviewer PdgB (3)**
> > >
> > > < Continued from Response to Reviewer PdgB (2) >
> > >
> > > - **Clarity**
> > >
> > >    - **Figure 1, (c) and (d):** We note that 1(c) and 1(d) are indeed the same method - FRR (Ours), 5-Rep. This represents the decision boundary obtained using the proposed FRR regularizer in the case of 5 feature replications along the y-axis. In (c) we show the train data, while in (d) we show the test or OOD data. (d) demonstrates that the proposed regularizer results in a max-margin classifier even in the presence of feature replications, and hence it is more robust to distribution shifts in data. We include the notes - (ID) or (OOD) in the updated submission for better clarity.
> > >
> > > - **Coloured MNIST**:
> > >    - **Metric used for correlation**: The analysis was performed on the test distribution, which combines randomly coloured backgrounds (sampled uniformly from a range of colors consisting of convex combinations of Red and Green) against different shapes of digits '1' and '5'. Pearson correlation coefficients are then computed between each individual feature (among the 32 features in the network) and the color (First channel was used, since it is also correlated to the second channel. The third channel is always 0) using the *corrcoef* function in numpy. We also compute the correlation coefficient between each feature and the shape (either '1' or '5'). Therefore, for each of the 32 features, we have two correlation values, corresponding to color and shape respectively. We now term a feature as "Color feature" if the absolute value of its correlation w.r.t. color is higher than that w.r.t. shape, and is also higher than 0.8 ($r^2 > 0.9$). Similar criteria is used to call a feature as a "Shape feature". Since there may be some features which do not satisfy either of the two criteria, the sum of shape and color features does not match with the total number of features in Table-1. We show the values of shape and color correlations in the below table for better clarity:
> > >
> > > | Feature Index | Absolute Correlation with Colour | Absolute Correlation with Shape | Feature Index | Absolute Correlation with Colour | Absolute Correlation with Shape |
> > > |----|-------------|-----------|--------|---------|---------|
> > > |             0 |                             0.95 |                            0.07 |            16 |                             0.43 |                            0.66 |
> > > |             1 |                             0.91 |                            0.06 |            17 |                             0.84 |                            0.44 |
> > > |             2 |                             0.93 |                            0.16 |            18 |                             0.34 |                            0.81 |
> > > |             3 |                             0.05 |                            0.91 |            19 |                             0.17 |                            0.68 |
> > > |             4 |                              0.9 |                            0.07 |            20 |                             0.92 |                            0.06 |
> > > |             5 |                             0.82 |                            0.22 |            21 |                             0.08 |                            0.92 |
> > > |             6 |                              0.7 |                            0.52 |            22 |                             0.95 |                            0.09 |
> > > |             7 |                             0.93 |                            0.07 |            23 |                                0 |                            0.91 |
> > > |             8 |                             0.81 |                            0.08 |            24 |                             0.91 |                            0.08 |
> > > |             9 |                             0.92 |                            0.07 |            25 |                              0.9 |                            0.22 |
> > > |            10 |                             0.91 |                            0.08 |            26 |                             0.94 |                            0.07 |
> > > |            11 |                             0.96 |                            0.07 |            27 |                             0.94 |                            0.17 |
> > > |            12 |                             0.94 |                            0.19 |            28 |                             0.92 |                            0.05 |
> > > |            13 |                             0.91 |                            0.17 |            29 |                             0.94 |                            0.07 |
> > > |            14 |                             0.86 |                            0.03 |            30 |                             0.72 |                            0.29 |
> > > |            15 |                             0.95 |                            0.07 |            31 |                             0.92 |                            0.07 |
> > >
> > > <Continued in Response to Reviewer PdgB (4) >

---

> > > > ### Author Response · Authors · 2022-11-11
> > > > **Response to Reviewer PdgB (4)**
> > > >
> > > > <Continued from Response to Reviewer PdgB (3) >
> > > >
> > > > - **More experiments need to be done such as on larger datasets and comparing with other SoTA methods:**
> > > >    We would like to highlight that the most widely used benchmark for Domain Generalization is DomainBed and consists of 5 Real World datasets with different types of distribution shifts. We present results on all of these datasets as is commonly done in other works [1,2,3]- PACS, VLCS, OfficeHome, Terra Incognita and DomainNet. Amongst these, DomainNet is a large scale dataset consisting of 6 domains, 345 classes and 0.6 million images. We compare with several SoTA methods in Table-3 of the main paper, and also Tables 7-11 in the Appendix. We compare even with MIRO, which was published very recently (October end) at ECCV 2022.
> > > > - **Clarification on Proposition 3.2:**
> > > >       As discussed in Section-F, we experiment with two implementations of $\ell_\infty$ norm: $\ell_{1,\infty}$, where we first compute the $\ell_\infty$ of feature reconstruction for each example in a batch and then average it across the batch, and $\ell_{\infty,1}$ where we compute the average $\ell_1$ reconstruction norm of each feature across the batch, and then apply $\ell_\infty$ norm on this $m$ dimensional vector. We consider the latter case in the proposition. We further replace the $\ell_1$ norm across the training minibatch with $\ell_2$ norm for ease of computation in the proof. We however note that empirically the results hold for both cases of $\ell_1$ and $\ell_2$ norm across the training minibatch, as we show on two datasets from DomainBed below. We use SWAD+FRR with the two different norm choices, and report the results. We apologize for the minor typo in the statement of the proposition which we have corrected in the updated draft.
> > > >
> > > >
> > > > | Norm     | OfficeHome | PACS |
> > > > |----------|------------|------|
> > > > | $\ell_1$ | 70.2       | 89.2 |
> > > > | $\ell_2$ | 70.4       | 88.9 |
> > > >
> > > > [1] Arpit et al., Ensemble of averages: Improving model selection and boosting performance in domain generalization
> > > >
> > > > [2] Cha et al., Swad: Domain generalization by seeking flat minima
> > > >
> > > > [3] Cha et al., Domain generalization by mutual-information regularization with pre-trained models.
> > > >
> > > > We thank the reviewer for the valuable comments and hope the reviewer would reconsider the contributions of our work in light of the clarifications and results presented in the rebuttal.

---

> > > > > ### Comment · Reviewer_PdgB · 2022-11-28
> > > > > **Updated rating**
> > > > >
> > > > > Thank you for the response. Some of my questions are answered but I still believe the paper's quality and clarity can be improved further. Therefore I would raise my score but still lean toward reject.

---

> > > > > > ### Author Response · Authors · 2022-11-30
> > > > > > **Requesting more details on the remaining concerns**
> > > > > >
> > > > > > We sincerely thank the reviewer for going through our rebuttal and updating their score.
> > > > > >
> > > > > > We would like to politely highlight that the other reviewers find our paper to be clear [Rne6, 4rL6] with a logical flow [GBf7]. Based on the constructive feedback from all reviewers we believe we have improved the quality and clarity of our draft further. We request the reviewer to kindly review our updated draft, and if possible, our responses to other reviewers as well, and kindly let us know of any further improvements which can help improve our work. We will be including updates from recent responses as well in the final version, which we are unable to do now since the revision upload window is now closed.
> > > > > >
> > > > > > We kindly request the reviewer to share more details on the remaining issues that were not addressed/ answered in the rebuttal. We will be happy to try our best to address them. Details on where the clarity/ quality can be improved would also be very helpful to us.

---

### Official Review · Reviewer_GBf7 · 2022-10-23

**Confidence:** 4
**Correctness:** 2
**Technical Novelty And Significance:** 2
**Empirical Novelty And Significance:** 2
**Recommendation:** 5

**Clarity, Quality, Novelty And Reproducibility:**

- Clarity
     - It is unclear to me how each of these major components are connected.
          - how does the thereory supports the learning of a non-simple features? it seems there are not even features that are simple or not formally defined in the theory?
          - it's also unclear to me how the proposed method is linked to the hypothesis or the theoretical discussion. The hypothesis or the theory does not seem to touch anything about "recoverable" to me.
          - the method talks about "recoverable", which is different from the wide-accepted term invertible as the title suggests.

- Quality
     - the theory discussion argues to connect the linear classifier to max-margin (SVM) classifier, but it seems to me the connection is a direct result by building the SVM loss function into the premise of the theory (the constraint in proposition 3.2)
     - the empirical scope does not seem to be comprehensive enough:
          - if the authors decide to use DomainBed, they need to follow the exact settings used in DomainBed, otherwise, they probably cannot just ignore all the methods that are shown worse than ERM by DomainBed
          - there is only one experiment other than this incomplete usage of DomainBed

- novelty
    - hard to evaluate due to the questions in clarity above.

- reproducibility:
     - good. replication of experiments are well performed.

- other:
     - the issues of learning the biased classifier from the data and countering it has been studied in multiple lines of works, I will recommend the authors to consider a more comprehensive summary of the methods in the first part of the related work section
         - Learning De-biased Representations with Biased Representations
         - Learning robust global representations by penalizing local predictive power
         - invariant risk minimization (it's already cited, but probably also worth some discussion in the related work part)

**Strength And Weaknesses:**

- strength
    - the flow of the paper (a hypothesis, validation, theoretical discussion, empirical results) goes very well, and logic of the contributions are expanded nicely.
    - the paper studies an important problem and build another layer to the current knowledge of the problem.

- weakness
   - how each pieces of the main logic flow are connected together is elusive to me (see detailed comments below)
   - the authors use DomainBed as an empirical testbed for the results, but not following the standard test protocol that DomainBed introduces: it seems there are missing datasets from the DomainBed protocol.


**Summary Of The Paper:**

The paper studies the problem of countering the existence of bias features in the data, with a hypothesis that the classifier learns the simple features (that are not generalizable in OOD) settings with too many weights, validating this hypothesis in simple MNIST experiment, and then offer some analytical discussions with correspondence to SVM, and compare to some SOTA line experiments. Each part of these above discussions seems reasonably good, but there seem some concerns when these pieces are combined into a coherent manuscript.

**Summary Of The Review:**

the overall paper is interesting, especially the several major components. However, the internal connection of these components seems missing.

---

> ### Author Response · Authors · 2022-11-11
> **Response to Reviewer GBf7 (1)**
>
> **A note to all reviewers:**
>
> We sincerely thank the reviewers for their time and valuable feedback on our work. At the outset, it is encouraging that all four reviewers find our observation on *Feature Replication Hypothesis (FRH)* interesting and novel, and some reviewers find our FRR regularizer easy to understand and implement, experiments promising, and writing to be clear with a logical flow.
>
> We have identified two common concerns in all reviews -
>
> - **Justification of FRH**: While we intended to show this as an assumption, and hence term this as a *Hypothesis*, we understand that since this is a key component and contribution of the paper, it needs better justification.
>
>  - **Coloured MNIST**: We note that a second common concern was the explanation of the Colored MNIST experiments.
>
> We justify these major concerns and the remaining points raised by each reviewer in our rebuttal.
>
> We believe the insightful feedback from all the reviewers has indeed helped us identify and strengthen the weak links in our work. We hope to have a fruitful discussion through the discussion period, which can be very helpful in making this work a valuable contribution to the research community. We have also updated the draft with our responses in appropriate sections. The updates in the draft have been done in blue for better clarity.
>
> **Comments to Reviewer *GBf7*:**
>
> - **Missing Datasets from DomainBed protocol**
>    - We would like to clarify that our goal of using DomainBed was to demonstrate the scalability of the proposed approach to Real-world datasets. We therefore evaluate on all Real-world datasets in DomainBed - PACS, VLCS, OfficeHome, TerraIncognita and DomainNet, which are the most common datasets used in literature as well [1,2,3].
>    - Since we show comprehensive results on the more complex synthetic (semi-real) dataset MNIST-CIFAR-10, we skip the two synthetic datasets on DomainNet - CMNIST and RMNIST.
>
>
> - **Clarity**
>
>   We present a brief summary to highlight the connection between the hypothesis, theory and the proposed approach. We also answer the questions posed by the reviewer in detail -
>
>     - [FRH] Based on observations from prior works, we put forth the (Simple) Feature Replication Hypothesis (FRH) which states that - Simplicity Bias is observed because the simpler features of the input are replicated multiple times in the feature space of neural networks.
>       - We term this as a hypothesis since we do not prove this theoretically.
>       - However, we explain the implication of the hypothesis in a practical setting and present a justification based on observations from prior works in Appendix-B of the revised submission.
>
>    - From prior work [5] we know that the implicit bias of SGD (and GD) causes models to converge to a max-margin or SVM solution.
>    - [Claim 3.1] We now show theoretically that in the presence of replicated features, an SVM classifier gives higher importance to these replicated features.
>       -  It is to be noted that we do not make claims on "simple" features in our theory, and hence we do not formally define features that are "simple".
>   - We now combine FRH and Claim-3.1 - We know from the Feature Replication Hypothesis (FRH) that simple features are replicated. We know from the theory that replicated features get more importance in an SVM classifier (which is closely related to the solution of SGD due to its implicit bias). This shows why Neural Networks latch on to simple features during training.
>   - The hypothesis and theory show why SGD is unable to use rare and complex features for better generalization, despite its implicit bias to latch on to the best possible (max-margin) solution. Our method FRR (Feature Reconstruction Regularizer) proposes a means to fix this, by enforcing that the network utilizes and remembers all features, ones that are rare also.
>   - Proposition-3.2 justifies how using FRR can result in a max-margin solution in the non-replicated feature space, which is the best possible solution for OOD generalization.
>
> - **Recoverable vs. Invertible**
>    - As discussed in the introduction, FRR enables the learning of an Invertible Mapping in the output layer, only for the domain of features seen during training.
>    - As highlighted by the reviewer, the title could be misleading when this explanation is ignored, hence we update it to the following in the revision - "Feature Reconstruction From Outputs Can Mitigate Simplicity Bias in Neural Networks"
>
> [1] Arpit et al.,  Ensemble  of  averages:   Improving  model  selection  and  boosting  performance  in  domain  generalization
>
> [2] Cha et al.,  Swad:  Domain generalization by seeking flat minima
>
> [3] Cha et al., Domain generalization by mutual-information regularization with pre-trained models.
>
> <Response continues in the next post, Response to Reviewer GBf7 (2)  >

---

> > ### Author Response · Authors · 2022-11-11
> > **Response to Reviewer GBf7 (2)**
> >
> > < Continued from Response to Reviewer GBf7 (1) >
> >
> > - **Connection with SVM classifier**
> >   - We use the SVM training formulation in Proposition 3.2 since it is known from prior works that the implicit bias of SGD makes networks converge to a max-margin solution.
> >   - We compare this against the standard SVM training formulation, which fails, as shown in Claim-3.1.
> >   - So we compare SVM+FRR (Ours) with SVM+L2 regularizer (baseline), which we believe is a fair comparison. Therefore, our conclusion is not based on forcing an SVM objective into the constraint of the proposition.
> >
> > - **Incomplete usage of DomainBed**
> >    - We indeed follow the same settings specified in the DomainBed paper, including the use of in-domain validation set for hyperparameter tuning.
> >    - We defer the results that are worse than ERM to Tables 7-11 in the Appendix due to space constraints in the main paper.
> >
> > - **There is only one experiment other than DomainBed**
> >   - We would like to highlight that DomainBed is itself a composition of several datasets. Prior works report results on all 5 real datasets of DomainBed [1,2,3], and we also report the same.
> >   - We also point the reviewer to the empirical justification of the Feature Replication Hypothesis (FRH) that has been added in Appendix-C of the revised submission. We consider 2 more synthetic datasets and one real dataset to justify the same.
> >
> > -  **Additional works**
> >     - We thank the reviewer for pointing to the additional papers. We have discussed them in the revised version.
> >
> > We thank the reviewer for the valuable comments and hope the reviewer would reconsider the contributions of our work in light of the clarifications and results presented in the rebuttal.

---

> > > ### Comment · Reviewer_GBf7 · 2022-12-07
> > > **response to rebuttals**
> > >
> > > Thanks for the clarifications, the revision of the title as a more actual reflection of the techniques helps a lot. I revised my ratings.

---

> > > > ### Author Response · Authors · 2022-12-07
> > > > **Requesting more details on the remaining concerns**
> > > >
> > > > We sincerely thank the reviewer for reviewing our rebuttal and updating the rating.
> > > > It would be very helpful if you could share the concerns that we were unable to address in our rebuttal. We will try our best to address the same in the remaining time.

---

> > > > > ### Comment · Reviewer_GBf7 · 2022-12-07
> > > > > **response to remaining concerns**
> > > > >
> > > > > Sure, it will be helpful if there is a direct response to the following remark in the original review.
> > > > >
> > > > > > the theory discussion argues to connect the linear classifier to max-margin (SVM) classifier, but it seems to me the connection is a direct result by building the SVM loss function into the premise of the theory (the constraint in proposition 3.2)
> > > > >
> > > > > There are some responses in the "Connection with SVM classifier" section, which reads like the authors agree with the remark and then mention more contributions in addition to this direct result.
> > > > >    - If this is indeed this case, which is totally fine (since there are more contributions as the authors mentioned), but then the writing of the paper should faithfully reflect such a case.
> > > > >    - If not, please clarify further.

---

> > > > > > ### Author Response · Authors · 2022-12-07
> > > > > > **Response to the remaining concerns**
> > > > > >
> > > > > > We thank the reviewer for the quick reply.
> > > > > >
> > > > > > We wish to clarify that our goal is not to connect the linear classifier to max-margin (SVM) classifier, since prior works [1] have already shed light on the same. In fact, we wanted to show why the classifier fails to generalize to distribution shifts despite learning the best possible (max-margin) classifier. More precisely, we aim to answer the following:
> > > > > > Although one line of prior art shows that ERM already learns diverse features, and another line shows that SGD/ GD training leads to a max-margin classifier, why is the classifier unable to utilize all possible features to maximize margin?
> > > > > >
> > > > > > We answer this using Claim 3.1 - SVM classifier has a tendency of being biased to replicated features and giving lesser importance to sparse features.
> > > > > >
> > > > > > In Prop.3.2, we use our regularizer as an alternative to the standard $l_2$ objective in the hard margin SVM formulation to show that this weakness of SVM classifiers can be overcome using the proposed regularizer. The resulting classifier gives equal importance to both replicated and sparse features.
> > > > > >
> > > > > > However, as rightly pointed out by the reviewer, our main loss does not contain the SVM term and we introduce this only in the theory. But the aim of doing this was not to build the connection between linear and SVM classifiers, which already existed in the prior literature. Rather, the aim was to use this connection to justify why our method is possibly working.
> > > > > >
> > > > > > While we are unable to update this clarification currently in the submission due to the current restrictions on editing the same, we will certainly include this important clarification to convey the link between our theory and implementation. We will cite the theoretical basis [1] in Section 3.3 to justify why we consider the max margin classifier, clarifying that past work has found SGD to converge to max-margin classifiers when data is linearly separable.
> > > > > >
> > > > > > We thank the reviewer once again for sharing the remaining concern and hope this clarifies the same. We will be happy to clarify any further concerns as well.
> > > > > >
> > > > > > [1] Soudry et al., The implicit bias of gradient descent on separable data, JMLR 2018

---

### Official Review · Reviewer_4rL6 · 2022-10-23

**Confidence:** 3
**Correctness:** 2
**Technical Novelty And Significance:** 3
**Empirical Novelty And Significance:** Not applicable
**Recommendation:** 6

**Clarity, Quality, Novelty And Reproducibility:**

Clarity: The submission is reasonably clear.

Quality: This is a subjective assessment, and I’d rate the quality as being average to mildly-above average.

Novelty: The hypothesis and proposed method appear to be novel.

Reproducibility: There seem to be sufficient details in the submission for meaningful reproducibility.

**Strength And Weaknesses:**

The feature replication hypothesis is novel and intriguing, and somewhat intuitive. The empirical results seem to indicate improved performance, suggesting this is an effective method on the whole.

While the hypothesis is interesting, I find the empirical validation to be quite weak. There is just one experiment demonstrating it (corresponding to one dataset, one network architecture, one round of training). In effect, the dataset introduces significant variation in the easier-to-learn colour-features (because of the use of ranges instead of monochromatic colouring, and colouring the entire background instead of just the digit) coupled with restriction of shape-variation by using only two digits, which perhaps naturally encourages a network to learn multiple features for different colour-buckets (to account for cases when the correlation does not hold). Of course, the invariant feature is still shape, and this mode of failure might well arise in real life datasets as well. However, this single, highly-contrived experiment does not really serve as a sufficiently reliable model of reality, in my view, and therefore does not really constitute as empirical validation of the hypothesis.

Since the proposed method results in longer training overall, what happens if the baselines are also trained for just as long? It seems likely from the results in Table 2 that ERM by itself might not improve much, but the other baselines (such as RSC) might.

There seems to be some inconsistencies in the aggregated results in Table 3 vs. the expanded numbers in the Appendix. For example,

 — PACS: Table 3 lists SMA as 87.5 while Table 6 lists it as 95.5

 — VLCS: Table 3 lists SMA as 78.2 while Table 7 lists it as 80.7

 — OfficeHome: Table 3 lists ERM as 66.5 while Table 8 lists it twice, once as 67.6; Table 3 lists SMA as 70.6 while Table 9 lists it as 82.0

 — TerraIncognita: Table 3 lists ERM as 46.1 while Table 9 lists it twice, once as 47.8; Table 3 lists SMA as 59.7 while Table 9 lists it as 59.7

 — DomainNet: Table 3 lists ERM as 40.9 while Table 9 lists it twice, once as 44.0; Table 3 lists SMA as 46.0 while Table 9 lists it as 60.0

 — MIRO numbers aren’t shown in the Appendix-tables

The SMA discrepancies in particular might invalidate the claim that a new state-of-the-art is set.

**Summary Of The Paper:**

The submission hypothesizes that one of the underlying reasons behind OOD failures related to the simplicity bias in neural networks is that the features learned tend to include several replicates of simple features, while complex features are not similarly replicated. This mechanism is claimed to lead to greater emphasis on simpler features, which is responsible for corresponding OOD failures. To prevent this, the submission proposes to add an invertibility criterion to the features-to-logits mapping — imposing invertibility would discourage the nonidentifiability implicit in replicated units in the feature dimensions. This regularizer is shown to result in improvements for OOD benchmarks.

**Summary Of The Review:**

While I am not really convinced by the empirical demonstration nor the hypothesis, it seems that the idea of imposing a form of identifiability on the final-layer mapping has a positive effect. It is not very clear to me why this is the case, but I am happy to treat this as an empirical discovery with positive consequences. My initial rating is borderline, while I wait for the authors to figure out the discrepancies in the tables — if it turns out that the results are indeed state-of-the-art, I shall revise my scores.

---

> ### Author Response · Authors · 2022-11-11
> **Response to Reviewer 4rL6 (1)**
>
> **A note to all reviewers:**
>
> We sincerely thank the reviewers for their time and valuable feedback on our work. At the outset, it is encouraging that all four reviewers find our observation on *Feature Replication Hypothesis (FRH)* interesting and novel, and some reviewers find our FRR regularizer easy to understand and implement, experiments promising, and writing to be clear with a logical flow.
>
> We have identified two common concerns in all reviews -
>
> - **Justification of FRH**: While we intended to show this as an assumption, and hence term this as a *Hypothesis*, we understand that since this is a key component and contribution of the paper, it needs better justification.
>
> - **Coloured MNIST**: We note that a second common concern was the explanation of the Colored MNIST experiments.
>
> We justify these major concerns and the remaining points raised by each reviewer in our rebuttal.
>
> We believe the insightful feedback from all the reviewers has indeed helped us identify and strengthen the weak links in our work. We hope to have a fruitful discussion through the discussion period, which can be very helpful in making this work a valuable contribution to the research community. We have also updated the draft with our responses in appropriate sections. The updates in the draft have been done in blue for better clarity.
>
> <Response continues in the next post, Response to Reviewer 4rL6 (2) >

---

> > ### Author Response · Authors · 2022-11-11
> > **Response to Reviewer 4rL6 (2)**
> >
> > < Continued from Response to Reviewer 4rL6 (1) >
> >
> > **Comments to Reviewer *4rL6*:**
> >
> > - **Empirical Justification of Feature Replication Hypothesis (FRH) on Semi-Real Datasets**:
> >     We present results on several variations of the Coloured MNIST dataset to justify the concerns raised by the reviewer.  Results for all datasets are reported in the table shown at the end of this post.
> >    - [Concern: Variation in colour, no variation in shape] **TwoColouredMNIST**: In this dataset, we use images of digits superimposed on either of the two colours- red or green. The difference from Coloured-MNIST is that we consider only two colours for the background, rather than a range. We notice extreme simplicity bias in this case, with the network learning 32 colour features and 0 shape features. This results in worse performance on the OOD dataset when compared to what we report in Table-1 (49.5% vs. 59.1%). We note that MNIST dataset already has a lot of variation in shape of a single digit, as opposed to a fixed color that has no variation. Hence, we did not introduce any additional factors of variation in shape in the original results presented in the paper.
> >    - [Concern: No variation in shape] **Coloured-MNIST-MultiDigit**: This is similar to the Coloured-MNIST dataset described in Section-3.1, with the exception that each of the classes is now composed of two digits. More specifically, the digits '1' and '7' and mapped to Class 0 and digits '5' and '8' are mapped to Class 1. We note that '1' and '5' are chosen from the original Coloured-MNIST dataset, while the second digit (e.g.'7') in each class is selected to be one that is similar to the first digit ('1') in the same class. This dataset is constructed specifically to show that the issue of Simplicity Bias and FRH exists even when there is higher variation in the shape feature, and is reported as ColouredMNIST-MultiDigit below. We see that while more shape features are learnt as compared to Coloured-MNIST, the network still  relies more on colour to make its decisions.
> >    - [Concern: Background coloured rather than digit] **Digit-Coloured-MNIST**: This is similar to the Coloured-MNIST dataset described in Section-3.1, with the exception that the digit is coloured rather than the background. This dataset is constructed specifically to show that the issue of SB and FRH exist even when the region that is coloured, which is the extent to which simple features exist in the image is much lesser, and is reported as DigitColouredMNIST below. Although this dataset also demonstrates the presence of SB, we note that the average correlation of features with shape is higher when compared to the above datasets.
> >    - [Concern: Highly contrived dataset, one network architecture] We present similar observations on the Real-World dataset Office-Home in the next section.
> >
> >      We note that in all the additional datasets considered, simpler features are represented more in the network while complex (hard-to-learn) features are sparse. This empirically justifies our hypothesis in Section-3.
> >
> > | Dataset                |   Number               |  Number             | Avg Correlation - |- with input           | Correlation - |   - with output                | ID acc | OOD acc |
> > |-------------------------|--------------------|---------------------|----------------------------|------------|-----------------------------|--------------------|--------|---------|
> > |                         | **Colour** | **Shape** | **Colour**                  | **Shape** | **Colour**         | **Shape** |        |         |
> > | ColouredMNIST           | 26                 | 4                   | 0.76                       | 0.26       | 0.81                        | 0.61               | 99.9   | 59.1    |
> > | TwoColouredMNIST        | 32                 | 0                   | 0.90                        | -          | 0.82                        | -                  | 99.9   | 49.5    |
> > | ColouredMNIST-MultDigit | 17                 | 7                   | 0.59                       | 0.32       | 0.76                        | 0.64               | 99.3   | 64.2    |
> > | DigitColouredMNIST      | 26                 | 3                   | 0.76                       | 0.36       | 0.79                        | 0.45                 |   99.9     |     62.1    |
> >
> > <Response continues in the next post, Response to Reviewer 4rL6 (3) >

---

> > > ### Author Response · Authors · 2022-11-11
> > > **Response to Reviewer 4rL6 (3)**
> > >
> > > < Continued from Response to Reviewer 4rL6 (2) >
> > >
> > > - **Empirical Justification of Feature Replication Hypothesis (FRH) on Real Datasets:** We justified FRH using only the toy dataset because it is a synthetic setting where we are aware of the good and spurious features, making it very easy to compute correlations. The same cannot be done in a real dataset since there are several good and spurious features. Moreover, we are not explicitly aware of the spurious features. However, we understand that demonstrating FRH on a real dataset is very important to strengthen our hypothesis. We thus attempt to show that Feature Replication is seen even in Real datasets using a different experimental setting as described below.
> > >
> > >   We attempt to demonstrate feature replication in a model trained with ERM on the *Real* domain of OfficeHome. We train a ResNet-50 on this domain, and perform PCA on the features learnt by this network. The network learns 2048 features per example, and we compute the $2048 \times 2048$ sized covariance matrix of the features over samples from a test domain (Clipart). We then compute the eigenvalues of this matrix, and find that 500 principal components can explain about 97.5\% of the variance, i.e. the matrix is extremely low rank, as shown in Fig.3 of the Appendix. This points to the fact that a lot of the learnt features are linearly dependent and highly correlated with each other. This trend is similar to what we observed on ColouredMNIST, where a large number of features were highly correlated with the colour, and in turn with each other.
> > >
> > > - **Baselines being trained for longer:**
> > >      +  The baselines have either been taken from the respective paper or from the popular DomainBed benchmark [1] which is a widely accepted standard for Domain Generalization. The number of training iterations are specific to each dataset, and chosen such that the training algorithm converges. Hence, we do not expect an increase in performance with an increase in training iterations.
> > >    + It is to be noted that all methods on DomainBed do not have the same training complexity, and yet they are all compared in the same table. For example, DANN [2] and CDANN [3] require the use of a discriminator which involves larger memory and training complexity. This shows that the overall goal is to compare the converged accuracy of different methods, rather than benchmarking all methods in a given compute budget.
> > >    +  Lastly, we would like to highlight that our method can be performed on a feature extractor trained with any base algorithm to obtain improvements. Thus, even if training on larger iterations (upto convergence) improves accuracy of any given baseline, we can obtain a further boost by using the proposed FRR-L and FRR-FLFT steps on top of it.
> > >      + While we are unable to report the accuracy of all baselines with a higher compute due to limited time, we show an example for the RSC algorithm below. We further note that the closest baseline that we need to be concerned with is SWAD. This algorithm in fact stops much before the total number of iterations finish, based on the detection of a basin/ convergence. Therefore training on a higher number of iterations (2 times the number of original iterations i.e. PACS, VLCS, and OfficeHome were now trained for 10000 iterations and DomainNet on 20000 iterations)  does not improve the accuracy as shown below. We note that the accuracy does not improve with higher training iterations as expected:
> > >
> > > | Algorithm | PACS | VLCS | OfficeHome | DomainNet |
> > > |-----------|------|------|------------|-----------|
> > > | SWAD (Original)      | 88.1 | 79.1 | 70.6       | 46.5      |
> > > | SWAD (Longer)      | 87.9 | 78.8 | 70.9       | 46.7      |
> > > | RSC  (Original)     | 85.2 | 77.1 |  65.5      | 39.1      |
> > > | RSC  (Longer)     | 84.8 | 76.9 | 65.7       | 39.1      |
> > >
> > > - **Inconsistencies in results:**
> > >   - We apologize for the error - the results in Tables 7-11 (Appendix) for SMA were a typo (they are for a different architecture). The reported results in the SMA paper on ResNet-50 correspond to the results in Table 3. We have corrected the tables in the Appendix in the revised submission.
> > >   - For ERM, the results in Table 3 are from the settings described in the github repo of domainbed, since this is what we build up on, and correspond to the earlier rows.
> > >   - For MIRO, the aggregated results are reported from their paper, and they do not have a complete domain wise breakdown in tables, but only have graphs. We re-run this baseline and add the results to the appendix tables in the revised submission.
> > >
> > > We thank the reviewer for the valuable comments, which have helped us strengthen our work.

---

### Official Review · Reviewer_Rne6 · 2022-10-25

**Confidence:** 4
**Correctness:** 2
**Technical Novelty And Significance:** 2
**Empirical Novelty And Significance:** 2
**Recommendation:** 6

**Clarity, Quality, Novelty And Reproducibility:**

The writing is definitely clear, and the method is reproducible for sure. The method and overall idea are fairly simple, and I'm unsure about the novelty, especially since it is just a regularization strategy that empirically seems to work well, but it is still not backed up well by theory or in-depth analysis of any kind.

**Strength And Weaknesses:**

# Pros:
1. Their Feature Replication Hypothesis is easy to understand and forms a good attempt to explain simplicity bias (SB)
2. Their Feature Reconstruction Regularizer (FRR) scheme to mitigate SB is also easy to understand and implement
3. Overall, the experiments seem promising

# Cons
1. The authors say that the main reason for SB is the implicit bias of SGD (more specifically, the stochastic component of it) to converge to max-margin solutions. I am not sure that this is the only main reason. Can the authors show an experiment using full-batch gradient descent (which should be easily possible on toy datasets used in the paper like MNIST/CIFAR, colored-MNIST etc) and its impact on SB? If SGD is the main cause for SB, then full-batch gradient descent should perform better in this regard. If full-batch gradient descent also shows SB, then it's questionable whether the feature replication hypothesis sufficiently explains the SB issue observed in DNNs.

2. I'm skeptical about the results and conclusions derived from Table-1. How the correlation analysis was performed is not clear. What is the metric used? The thing that bothers me in Table-1 is there is no reason to believe features in the last layer are disentangled wrt color and shape. There is no reason to assume that color and shape will be disentangled by the feature extractor for free (without any explicit constraints to ensure this). Can you better justify why features must be disentangled here? Apart from this toyish setup, do you have any more concrete realistic datasets/experiments/results to support the feature replication hypothesis? This is concerning as its one of the main
points of the paper, and it's not well justified according to me.

3. Another concern is why Eq-1 in the paper (L_FRR) regularizer results in diverse features. The authors do not provide much reasoning or justification in this regard. Intuitively it appears that this regularizer is trying to enforce the full rank of the W linear weight matrix. I suggest authors explain this in more detail to help the readers understand this. And it's also not clear why the explicit enforcement of a full-rank constraint doesn't work. The authors say its a very stringent constraint, but their regularizer also is trying to learn a transformation
that inverts the W matrix, which is essentially trying to enforce the full rank of W. Some more clarity is needed in this regard; why is there such a stark difference between these two seemingly similar strategies?


4. The authors should mention the effect of lambda in Eq-4,5. An ablation study in this regard will help know the sensitivity of the method wrt lambda. How dependent is the method on the choice of this value, and what is the possible range of suitable values for this hyperparameter? Maybe I missed this point in the main paper. If this is already present, please point me to it.

5. I think there is a lack of in-depth analysis of the cause and reasoning for the simplicity bias and the other observations the authors make in the paper. And, in the end, the authors simply seem to propose a regularization strategy. The results seem promising, but I feel it is not well supported or backed by in-depth analysis or theory, and hence I'm not fully convinced of the overall story being portrayed.

**Summary Of The Paper:**

The paper tries to shed light on the cause and nature of simplicity bias and proposes a regularization strategy to mitigate the same.
While we know that deep neural networks (DNNs) are prone to learning simple features (simplicity bias), recent work provides evidence
that the penultimate features in the DNNs contain a diverse and complex set of features.  The authors try to bridge the gap between these seemingly contradictory observations. They try to show that the features learned by the DNN are highly biased toward simple features. The simple features are replicated to a large degree causing the SGD to converge to max-margin solutions (due to its implicit bias) that primarily rely on simple features. The authors further devise a regularization scheme (FRR) to mitigate simplicity bias. They use FRR-L to retrain only the last layer (while freezing the backbone), and this is followed by retraining the backbone while freezing the last linear layer. This approach enforces the model to learn a rich and diverse set of features, leading to better OOD generalization. They show SOTA results on standard OOD benchmark datasets.

**Summary Of The Review:**

The idea of the paper is to impose a regularization scheme to somehow alleviate SB, but why their method works is not very clear, and some of the claims, like how the stochasticity part of SGD (which is what leads to max-margin solutions) is responsible for SB. And are the simple features actually replicated a lot? The simple colored-MNIST eg in the table-1 is dissatisfactory as the authors assume that the backbone or DNN feature extractor disentangles shape and color for free during training. A single neuron in the feature layer could be correlated with both color and shape. How do the authors justify that each neuron is majorly responsible only for a single attribute (color or shape). Even if this holds true for colored-MNIST, it is seldom the case in more realistic datasets of high resolution like ImageNet, CelebA or AFHQ (especially since there is no explicit enforcement of any such disentanglement constraint). So the core hypothesis in the paper, which is feature replication, itself is questionable and not fully justified. I feel that the paper lacks an in-depth analysis of the problem, and certain observations and claims seem superficial and maybe not fully true. The authors should better justify all this. I'm open to changing my review after the rebuttal if the authors convince me of the overall story of the paper.

---

> ### Author Response · Authors · 2022-11-10
> **Response to Reviewer Rne6**
>
> **A note to all reviewers:**
>
> We sincerely thank the reviewers for their time and valuable feedback on our work. At the outset, it is encouraging that all four reviewers find our observation on *Feature Replication Hypothesis (FRH)* interesting and novel, and some reviewers find our FRR regularizer easy to understand and implement, experiments promising, and writing to be clear with a logical flow.
>
> We have identified two common concerns in all reviews -
>
> - **Justification of FRH**: While we intended to show this as an assumption, and hence term this as a *Hypothesis*, we understand that since this is a key component and contribution of the paper, it needs better justification.
>
> - **Coloured MNIST**: We note that a second common concern was the explanation of the Colored MNIST experiments.
>
> We justify these major concerns and the remaining points raised by each reviewer in our rebuttal.
>
> We believe the insightful feedback from all the reviewers has indeed helped us identify and strengthen the weak links in our work. We hope to have a fruitful discussion through the discussion period, which can be very helpful in making this work a valuable contribution to the research community. We have also updated the draft with our responses in appropriate sections. The updates in the draft have been done in blue for better clarity.
>
> **Comments to Reviewer *Rne6*:**
>
>
> - **Relation between SGD/GD and Simplicity Bias + Overall Story**: We would like to clarify that we do not claim that the implicit bias of SGD is the main reason for SB. In fact we try to understand why SB exists despite the implicit bias of SGD to converge to a max-margin solution, which is possibly the best possible solution with this loss. We clarify our line of thought below -
>
>    Despite the existence of several optimal solutions that minimize training loss in Neural Networks, SGD is still able to converge to one that generalizes well. Prior works [1] attribute this property to the implicit bias of SGD (and GD) to converge to a max-margin solution when compared to any solution that minimizes train loss. Another line of work also shows that ERM training indeed learns diverse features [2,3], but the classifier does not utilize all of them in ERM training. We try to understand why this happens despite the implicit bias of SGD (and GD) to converge to a max-margin solution, which should have ideally considered all possible features to maximize margin. We show that max-margin classifiers fail to use rare features in the presence of repeated features (Claim 3.1). To ensure that the model considers these rare features for classification, we impose the Feature reconstruction regularizer - FRR. If the model does not utilize the rare features, it cannot reconstruct them, resulting in a higher FRR loss. We also show theoretically that the proposed regularizer leads to the learning of a max-margin classifier in the non-replicated feature space.
>
>    We therefore attribute SB to the existence of replicated simple features, and rare complex features. We would like to note that Soudry et al. [1] show the existence of Implicit Bias for Full Batch Gradient Descent as well.
>
> [1] Soudry et al., The implicit bias of gradient descent on separable data, JMLR 2018
>
> [2] Rosenfeld et al., Domain-adjusted regression or: Erm may already learn features sufficient for out-of-distribution generalization, 2022
>
> [3] Kirichenko et al., Last layer re-training is sufficient for robustness to spurious correlations, 2022

---

> > ### Author Response · Authors · 2022-11-10
> > **Response to Reviewer Rne6 (2)**
> >
> > - **Coloured MNIST**:
> >    - **Metric used for correlation**: The analysis was performed on the test distribution, which combines randomly coloured backgrounds (sampled uniformly from a range of colors consisting of convex combinations of Red and Green) against different shapes of digits '1' and '5'. Pearson correlation coefficients are then computed between each individual feature (among the 32 features in the network) and the color (First channel was used, since it is also correlated to the second channel. The third channel is always 0) using the *corrcoef* function in numpy. We also compute the correlation coefficient between each feature and the shape (either '1' or '5'). Therefore, for each of the 32 features, we have two correlation values, corresponding to color and shape respectively. We now term a feature as "Color feature" if the absolute value of its correlation w.r.t. color is higher than that w.r.t. shape, and is also higher than 0.8 ($r^2 > 0.9$). Similar criteria is used to call a feature as a "Shape feature". Since there may be some features which do not satisfy either of the two criteria, the sum of shape and color features does not match with the total number of features in Table-1. We show the values of shape and color correlations in the below table:
> >
> > | Feature Index | Absolute Correlation with Colour | Absolute Correlation with Shape | Feature Index | Absolute Correlation with Colour | Absolute Correlation with Shape |
> > |---------------|----------------------------------|---------------------------------|---------------|----------------------------------|---------------------------------|
> > |             0 |                             0.95 |                            0.07 |            16 |                             0.43 |                            0.66 |
> > |             1 |                             0.91 |                            0.06 |            17 |                             0.84 |                            0.44 |
> > |             2 |                             0.93 |                            0.16 |            18 |                             0.34 |                            0.81 |
> > |             3 |                             0.05 |                            0.91 |            19 |                             0.17 |                            0.68 |
> > |             4 |                              0.9 |                            0.07 |            20 |                             0.92 |                            0.06 |
> > |             5 |                             0.82 |                            0.22 |            21 |                             0.08 |                            0.92 |
> > |             6 |                              0.7 |                            0.52 |            22 |                             0.95 |                            0.09 |
> > |             7 |                             0.93 |                            0.07 |            23 |                                0 |                            0.91 |
> > |             8 |                             0.81 |                            0.08 |            24 |                             0.91 |                            0.08 |
> > |             9 |                             0.92 |                            0.07 |            25 |                              0.9 |                            0.22 |
> > |            10 |                             0.91 |                            0.08 |            26 |                             0.94 |                            0.07 |
> > |            11 |                             0.96 |                            0.07 |            27 |                             0.94 |                            0.17 |
> > |            12 |                             0.94 |                            0.19 |            28 |                             0.92 |                            0.05 |
> > |            13 |                             0.91 |                            0.17 |            29 |                             0.94 |                            0.07 |
> > |            14 |                             0.86 |                            0.03 |            30 |                             0.72 |                            0.29 |
> > |            15 |                             0.95 |                            0.07 |            31 |                             0.92 |                            0.07 |
> >
> >    - **Disentanglement of features:** As rightly pointed by the reviewer, we can see from the above table that color and shape features are not disentangled. Categorizing each feature as "Color" or "Shape", and presenting the total number of Color and Shape features helps us understand that "Color" is more predominant than "Shape" in majority of the features. We additionally present the average correlation coefficient of both Color and Shape (which are 0.76 and	0.26 respectively) in Table-1 of the revised submission since this shows that overall correlation is also higher for Color when compared to Shape.

---

> > > ### Author Response · Authors · 2022-11-10
> > > **Response to Reviewer Rne6 (3)**
> > >
> > > **Formal Proof of FRH:** We do not present a theoretical proof of FRH, and hence refer to it as a *Hypothesis*. However, for better understanding, we state our intuition more formally below -
> > >
> > > In a practical scenario where features are not disentangled, our hypothesis translates to the following:
> > >
> > >   **Conjecture:** Simpler features of the input are represented more in the feature space of neural networks, while complex (hard-to-learn) features are sparse.
> > >
> > >   **Assumptions:**
> > >   -  We consider simple features such as background to be spurious, and complex features such as shape to be robust.
> > >   -  We consider an overparameterized network that has the capacity to learn more features than what exist, resulting in feature repetition.
> > >
> > >   **Justification:** We justify the conjecture by showing that all other possibilities discussed below cannot be true.
> > >
> > >   1. Assumption: DNNs learn only Simple Features
> > >
> > >      Contradiction: Prior works [2,3] show that features learned by ERM are diverse, and last layer training on target domain is good enough to obtain robustness to spurious features. This cannot be possible if the network has learned only spurious features.
> > >
> > >   2. Assumption: DNNs learn only Complex Features
> > >
> > >      Contradiction: The dominance of Simple features in the learning of DNNs is shown by Shah et al. [5]. Moreover, the existence of texture-bias [6] and background-bias [7] have been demonstrated in prior works, which show the dominance of Simple features.
> > >   3. Assumption: DNNs learn a uniform distribution of both Simple and Complex Features.
> > >
> > >       Contradiction: SGD converges to a max-margin solution due to its implicit bias. From Claim-3.1 (1), in the presence of balanced features that are correlated with the label, max-margin solution gives equal weight to all features. This contradicts the existence of Simplicity Bias [5].
> > >
> > >   4. Assumption: DNNs learn more Complex Features and less Simple Features.
> > >
> > >      Contradiction: Since Complex features are indeed more robust and are better correlated with the labels, the classifier would rely more on these features. This contradicts the existence of Simplicity Bias [5].
> > >
> > >   Therefore, the only feasible option which supports the empirical observations in prior works [1,2,3,4,5,6,7] is that DNNs learn more Simple Features and Complex features are sparse, which justifies our conjecture.
> > >
> > > - **Empirical Justification of Feature Replication Hypothesis (FRH) on Real Datasets:** We justified FRH using only the toy dataset because it is a synthetic setting where we are aware of the good and spurious features, making it very easy to compute correlations. The same cannot be done in a real dataset since there are several good and spurious features. Moreover, we are not explicitly aware of the spurious features. However, we understand that demonstrating FRH on a real dataset is very important to strengthen our hypothesis. We thus attempt to show that Feature Replication is seen even in Real datasets using a different experimental setting as described below.
> > >
> > >   We attempt to demonstrate feature replication in a model trained with ERM on the *Real* domain of OfficeHome. We train a ResNet-50 on this domain, and perform PCA on the features learnt by this network. The network learns 2048 features per example, and we compute the $2048 \times 2048$ sized covariance matrix of the features over samples from a test domain (Clipart). We then compute the eigenvalues of this matrix, and find that 500 principal components can explain about 97.5\% of the variance, i.e. the matrix is extremely low rank, as shown in Fig.3 of the Appendix. This points to the fact that a lot of the learnt features are linearly dependent and highly correlated with each other. This trend is similar to what we observed on ColouredMNIST, where a large number of features were highly correlated with the colour, and in turn with each other.
> > >
> > > We demonstrate our hypothesis on several other synthetic datasets in Appendix-C of the updated submission.
> > >
> > > [1] Soudry et al., The implicit bias of gradient descent on separable data, JMLR 2018
> > >
> > > [2] Rosenfeld et al., Domain-adjusted regression or: Erm may already learn features sufficient for out-of-distribution generalization, 2022
> > >
> > > [3] Kirichenko et al., Last layer re-training is sufficient for robustness to spurious correlations, 2022
> > >
> > > [4] Gulrajani et al., In search of lost domain generalization, 2020
> > >
> > > [5] Shah et al., The pitfalls of simplicity bias in neural networks, NeurIPS 2020
> > >
> > > [6] Geirhos et al., ImageNet-trained CNNs are biased towards texture; increasing shape bias improves accuracy and robustness, ICLR 2019
> > >
> > > [7] Xiao et al., Noise or Signal: The Role of Image Backgrounds in Object Recognition, ICLR 2021

---

> > > > ### Author Response · Authors · 2022-11-10
> > > > **Response to Reviewer Rne6 (4)**
> > > >
> > > > - **Why the proposed Feature Reconstruction Regularizer (FRR) works better than a Full rank constraint:**
> > > >   -  The proposed regularizer FRR attempts to reconstruct all features back from the logits, which ensures that the final layer does not ignore rare complex features.
> > > >   -   A  full rank solution $\widetilde{w}$ is one among the many $w \in \mathcal{W}$  solutions that satisfy this reconstruction criteria for the domain of inputs in the train set. The subspace of full rank solutions $\widetilde{\mathcal{W}}$ is a very sparse subset of $\mathcal{W}$. Let $w_{min}$ and $\widetilde{w}_{min}$
> > > >
> > > >       denote the optimal solutions of classification loss among the family of solutions $\mathcal{W}$ and $\widetilde{\mathcal{W}}$ respectively. Since $\widetilde{\mathcal{W}} \subset \mathcal{W}$, the optimal test set loss $L_{w_{min}} \leq L_{\widetilde{w}_{min}}$, showing that the proposed regularizer allows the existence of a better or similar solution when compared to imposing a full rank constraint, while also ensuring that sparse features are used. In practice, since the rank of inputs is very low in the presence of replicated features, the proposed regularizer obtains a significantly better solution when compared to a full rank regularizer. Further, we argue that for cases where the number of classes $k$ is much smaller than the number of features $d$, a full rank weight matrix could still end up using only $d$ features, and not necessarily be optimal for OOD generalization.
> > > >
> > > >   - We wish to highlight from Table-2 (E7 vs. E8) that the variance with the Full rank regularizer is $3\times$ higher than the proposed approach showing that this constraint is very hard and impacts the stability of training. As we increase the weight of the Full rank regularizer, training becomes more unstable as indicated by the higher variance (in the CIFAR-AvgMNIST accuracy) across runs in the below table -
> > > >
> > > >
> > > > |      | Accuracy (Mean) | Accuracy (Variance) |
> > > > |:----:|:---------------:|:-------------------:|
> > > > |   0  |      52.73      |         0.08        |
> > > > |  50  |      52.75      |         0.32        |
> > > > |  100 |      52.65      |         0.32        |
> > > > |  150 |      52.87      |         1.09        |
> > > > |  200 |      50.52      |         2.23        |
> > > > |  300 |      49.96      |         1.94        |
> > > > |  400 |      51.52      |         3.47        |
> > > > |  500 |      49.80      |         2.24        |
> > > > |  600 |      47.09      |         8.18        |
> > > > |  700 |      49.43      |         1.66        |
> > > > |  800 |      48.51      |         3.63        |
> > > > |  900 |      45.22      |         8.02        |
> > > > | 1000 |      14.04      |         7.66        |
> > > >
> > > >
> > > > -  **Effect of $\lambda$ and range of hyperparameters**
> > > >    - We present the range of hyperparameters in Table-5 of the supplementary.
> > > >    - We include the plot of accuracy against variation in hyperparameters in Fig.6 (Appendix) of the revised submission. It can be noted that the proposed method works across a wide range of $\lambda$ values for most domains.
> > > >
> > > > We thank the reviewer for the valuable comments, which have helped us strengthen our work.

---

### Decision · Program_Chairs · 2023-01-20

**Decision:**

Accept: poster

**Justification For Why Not Higher Score:**

The paper doesn't sound exciting enough to be a spotlight, as indicated by the reviewers' scores.

**Justification For Why Not Lower Score:**

Please see the summary of the strength.

The AC finds that the concerns of the reviewers with a rating 5 seem to be addressed by the authors (even though the reviewers didn't respond to the authors.)

**Metareview: Summary, Strengths And Weaknesses:**

As indicated by the reviewers, the paper has the following strength

"--- Their Feature Replication Hypothesis is easy to understand and forms a good attempt to explain simplicity bias (SB)
--- Their Feature Reconstruction Regularizer (FRR) scheme to mitigate SB is also easy to understand and implement
--- Overall, the experiments seem promising."



**Note From Pc:**

if the above contains the word "oral" or "spotlight" please see: "oral" presentation means -> notable-top-5% and "spotlight" means -> notable-top-25%. As stated in our emails, we are disassociating presentation type from AC recommendations